# Histone demethylase KDM2A is a selective vulnerability of cancers relying on alternative telomere maintenance

Fei Li[1,2,9], Yizhe Wang [3,9], Inah Hwang [3,4,9], Ja-Young Jang[3], Libo Xu[2,5], Zhong Deng[6], Eun Young Yu[7], Yiming Cai[8], Caizhi Wu [2], Zhenbo Han[8], Yu-Han Huang[2], Xiangao Huang[3], Ling Zhang[2,5], Jun Yao[8], Neal F. Lue [7], Paul M. Lieberman [6], Haoqiang Ying [8], Jihye Paik [3] ✉ & Hongwu Zheng [3] ✉

Telomere length maintenance is essential for cellular immortalization and tumorigenesis. 5% − 10% of human cancers rely on a recombination-based mechanism termed alternative lengthening of telomeres (ALT) to sustain their replicative immortality, yet there are currently no targeted therapies. Through CRISPR/Cas9-based genetic screens in an ALT-immortalized isogenic cellular model, here we identify histone lysine demethylase KDM2A as a molecular vulnerability selectively for cells contingent on ALT-dependent telomere maintenance. Mechanistically, we demonstrate that KDM2A is required for dissolution of the ALT-specific telomere clusters following recombination-directed telomere DNA synthesis. We show that KDM2A promotes de-clustering of ALT multitelomeres through facilitating isopeptidase SENP6-mediated SUMO deconjugation at telomeres. Inactivation of KDM2A or SENP6 impairs post-recombination telomere de-SUMOylation and thus dissolution of ALT telomere clusters, leading to gross chromosome missegregation and mitotic cell death. These findings together establish KDM2A as a selective molecular vulnerability and a promising drug target for ALT-dependent cancers.

Telomeres are specialized nucleoprotein structures that shield the linear chromosome ends of eukaryotes from promiscuous DNA repair and nucleolytic degradation activities[1]. Due to the chromosome "end-replication" problem, telomere DNA undergoes progressive attrition with each cell division[2]. Consequentially, proliferative tumor cells necessitate counteracting activity to maintain adequate telomere length and sustain their replicative immortality. While a majority of human cancers achieve this through telomerase activation, the remaining 5–10% of them rely on a homologous recombination-based mechanism termed alternative lengthening of telomeres (ALT)[3,4]. Recent studies further reveal ALT as a conservative DNA damage repair pathway analogous to break-induced replication (BIR) in budding yeast[5–8]. But the molecular pathway(s) that control ALT activation and termination still remain largely unclear.

[1]Department of Neurosurgery, Southwest Hospital, Chongqing 400038, China. [2]Cold Spring Harbor Laboratory, Cold Spring Harbor, NY 11724, USA. [3]Department of Pathology and Laboratory Medicine, Weill Cornell Medicine, New York, NY 10065, USA. [4]Graduate School of Pharmaceutical Sciences, College of Pharmacy, Ewha Womans University, Seoul 03760, Republic of Korea. [5]Key Laboratory of Pathobiology, Ministry of Education, and Department of Pathophysiology, College of Basic Medical Sciences, Jilin University, Changchun, Jilin, China. [6]The Wistar Institute, Philadelphia, PA 19104, USA. [7]Department of Microbiology and Immunology, W. R. Hearst Microbiology Research Center, Weill Cornell Medicine, New York, NY 10065, USA. [8]Department of Molecular and Cellular Oncology, The University of Texas M. D. Anderson Cancer Center, Houston, TX 77030, USA. [9]These authors contributed equally: Fei Li, Yizhe Wang, Inah Hwang. ✉e-mail: jep2025@med.cornell.edu; hoz4001@med.cornell.edu

In human cancers, ALT activation is intimately linked to the mutational status of the chromatin modulator genes *ATRX* and *DAXX*[9–13]. Functionally, ATRX and DAXX form a histone H3.3-specific chaperone complex that facilitates replication-independent nucleosome assembly at heterochromatic regions, including telomeres[14–18]. A survey of ~7000 patient samples in 31 cancer types found that 5% of them harbor genetic alterations of *ATRX* or *DAXX* that also concurrently present ALT features[19]. This tight association raises the possibility that ALT activation is a consequence of histone management dysfunction.

The ALT mechanism relies on homologous recombination-directed telomere DNA synthesis. Cumulative evidences suggest that ALT activation emanates from the telomere replication stress and the stalled replication forks[5,20,21]. Indeed, depletion of ATRX or DAXX disrupts replication-independent nucleosome incorporation and induces telomere chromatin de-condensation that progressively activates the recombination-directed telomere repair pathway[22]. As a consequence, the homologous repair-based ALT mechanism becomes the only viable path for the *ATRX* or *DAXX* mutant cells to achieve replicative immortality. In this sense, *ATRX* or *DAXX* loss, while promoting tumorigenesis by activating the ALT-directed telomere maintenance pathway, also simultaneously creates an intrinsic telomere replication defect that can potentially be exploited for synthetic lethal-like interactions.

Through CRISPR/Cas9-based genetic screens of isogenic ALT- and paired TERT-immortalized cell lines, here we identify histone demethylase KDM2A as a selective molecular vulnerability of cells that depend on ALT-directed telomere maintenance. We demonstrate that KDM2A functions to facilitate the dissolution of the ALT-specific multitelomere clusters following recombination-directed telomere synthesis. We further show that KDM2A promotes ALT multitelomere de-clustering by facilitating isopeptidase SENP6-mediated SUMO deconjugation at telomeres. These findings together establish KDM2A as a promising therapeutic target for ALT-dependent cancers.

## Results

### A CRISPR-based genetic screen of chromatin regulators required by ALT cells

To uncover the molecular vulnerabilities of cells that rely on ALT-directed telomere maintenance, we developed isogenic pairs of ALT cell lines from ATRX-depleted human lung IMR90 fibroblasts using our established immortalization protocol (Fig. 1a)[22]. Compared to the control IMR90-T lines that were immortalized by telomerase (TERT) expression, the ALT-immortalized IMR90 cells exhibited characteristically high levels of 53BP1-associated telomere dysfunction-induced foci (TIF) (Supplementary Fig. 1a, b), consistent with the notion that ALT telomeres experience chronic replication stress and are intrinsically unstable[5,12,23,24].

To profile chromatin regulators that are selectively required for ALT-immortalized cells, we constructed a library that contained ~5000 sgRNAs targeting 455 chromatin modifiers, readers, and effectors (~10 sgRNAs/gene) as well as ~100 control sgRNAs. To enhance the targeting efficiency, the sgRNAs were designed using an algorithm linked to protein domain annotation. The genetic screens were conducted by transducing the sgRNA library into three independently established ALT-immortalized IMR90 cells (hereafter referred to as ALT#1, ALT#2, and ALT#3) and two paired TERT-immortalized IMR90-T cells (referred to as IMR90-T#1 and #2). The pools of library-transduced cells were passaged for 16 population doublings before being subjected to next-generation sequencing-based quantification. The relative effect of each sgRNA on cell growth was scored by calculating the $\log_2$ fold-change (log2FC) of sgRNA abundances at the beginning and end of the culture periods (Fig. 1b and Supplementary Fig. 2a). The spike-in positive (sgPCNA, sgRPA3, sgCDK1, sgCDK9 sgTIP60, and sgTTF2) and non-targeting negative

(sgNeg1 – 100) control sgRNAs served as quality controls to validate the overall accuracy of the screening strategy.

For each gene, we calculated the gene dependency score (GDS) by averaging the $\log_2$FC of its targeting sgRNAs (4–14 per gene) (Supplementary Data 1). To rank the priority of ALT-selective vulnerabilities, the GDS of individual genes in indicated ALT cell lines (x-axis) were plotted against their scores in control IMR90-T cells (y-axis) (Fig. 1c). As expected, many of the identified gene dependencies were pan-essential and scored comparably in the ALT and control IMR90-T cells. Among those selectively required for ALT cells were several genes with diversified chromatin regulatory functions, including *ASM2L*, *KDM2A*, *KMT5B*, *RNF8*, and *SETDB1* (Fig. 1d). The most prominent hit in this screen was *KDM2A*, a member of the Jumonji C (Jmjc) domain-containing histone lysine demethylase family that targets lower methylation states of H3K36 (Kme1 and Kme2) but has no known telomere functions[25,26]. Notably, the majority (7 of 12) of sgRNAs targeting *KDM2A* included in the library screens caused robust growth inhibition phenotypes in all three ALT lines ($\log_2$FC <−5.0) but were ineffective against the paired IMR90-T control cells (Supplementary Fig. 2b), suggesting that KDM2A is selectively essential for ALT cells.

To validate the pooled screens, we next analyzed the growth impact of sgRNA targeting through fluorescence-activated cell sorting (FACS)-based competition assays[22]. Consistent with the screen result, targeting *KDM2A* using two newly designed sgRNAs (referred to as sgK#1 and sg#2) profoundly suppressed the growth of ALT-positive ALT#1 (Fig. 1e), osteosarcoma Saos2 (Fig. 1f), U2OS (Supplementary Fig. 2c), G292 (Supplementary Fig. 2d), rhabdomyosarcoma Hs729 (Supplementary Fig. 2e), and patient-derived pGBM6 glioblastoma cells (Supplementary Fig. 2f). As a complementary approach, we also conducted crystal violet-based clonogenic growth assays. As expected, sgK#1 or sgK#2-mediated KDM2A ablation in ALT#1 (Fig. 1g, h), Saos2 (Fig. 1i, j), or U2OS cells (Supplementary Fig. 2g, h) greatly inhibited their clonogenic growth. Moreover, the competition- and clonogenic-based proliferation assays demonstrated that complementation of the sgK#1- or sgK#2-resistant *KDM2A* cDNAs could fully rescue the growth inhibition caused by the respective sgRNAs (Fig. 1k and Supplementary Fig. 2i–l), confirming their on-target effect.

Finally, to assess whether KDM2A is essential for in vivo ALT tumor cell growth, Saos2 cells transduced with sgCtrl or *KDM2A*-specific sgRNAs were subcutaneously grafted into immunocompromised recipient mice. Analysis of tumor growth revealed that inactivation of KDM2A by sgK#1 or sgK#2 significantly inhibited the in vivo tumorigenicity (Fig. 1l, m), indicating that KDM2A is required for ALT-driven tumor propagation.

### KDM2A is not essential for non-ALT cell growth and survival

To ascertain whether KDM2A is selectively essential for ALT cells, we next examined the growth effect of KDM2A depletion on the wild-type *ATRX* cDNA-complemented U2OS cells. In line with the previous study[27], re-expression of ATRX in the ATRX-null U2OS cells suppressed ALT and APB formation (Supplementary Fig. 3a–c). Compared to the parental U2OS cells, targeting the ATRX-complemented U2OS with sgK#1 or sgK#2 caused a much-attenuated growth inhibition phenotype (Supplementary Fig. 3d), indicating that KDM2A is selectively essential for ALT cells. Consistently, ectopic expression of wild-type *DAXX* cDNA in the *DAXX*-mutant G292 cells suppressed the APB formation and rescued the growth inhibition caused by *KDM2A*-specific sgRNAs (Supplementary Fig. 3e–h)[28,29].

To further assess the selectivity of the KDM2A dependency, we next conducted competition-based proliferation assays in a panel of non-ALT cells of diversified tissue origins. Compared to ALT-positive cells in which ablation of KDM2A by sgK#1 or #2 led to 35–100-fold dropout in a period of 4 weeks, targeting *KDM2A* by the same sgRNAs incurred <2.5-fold growth inhibition in the panel of non-ALT human cell lines, including IMR90-T (Fig. 2a), HeLa of cervical cancer (Fig. 2b),

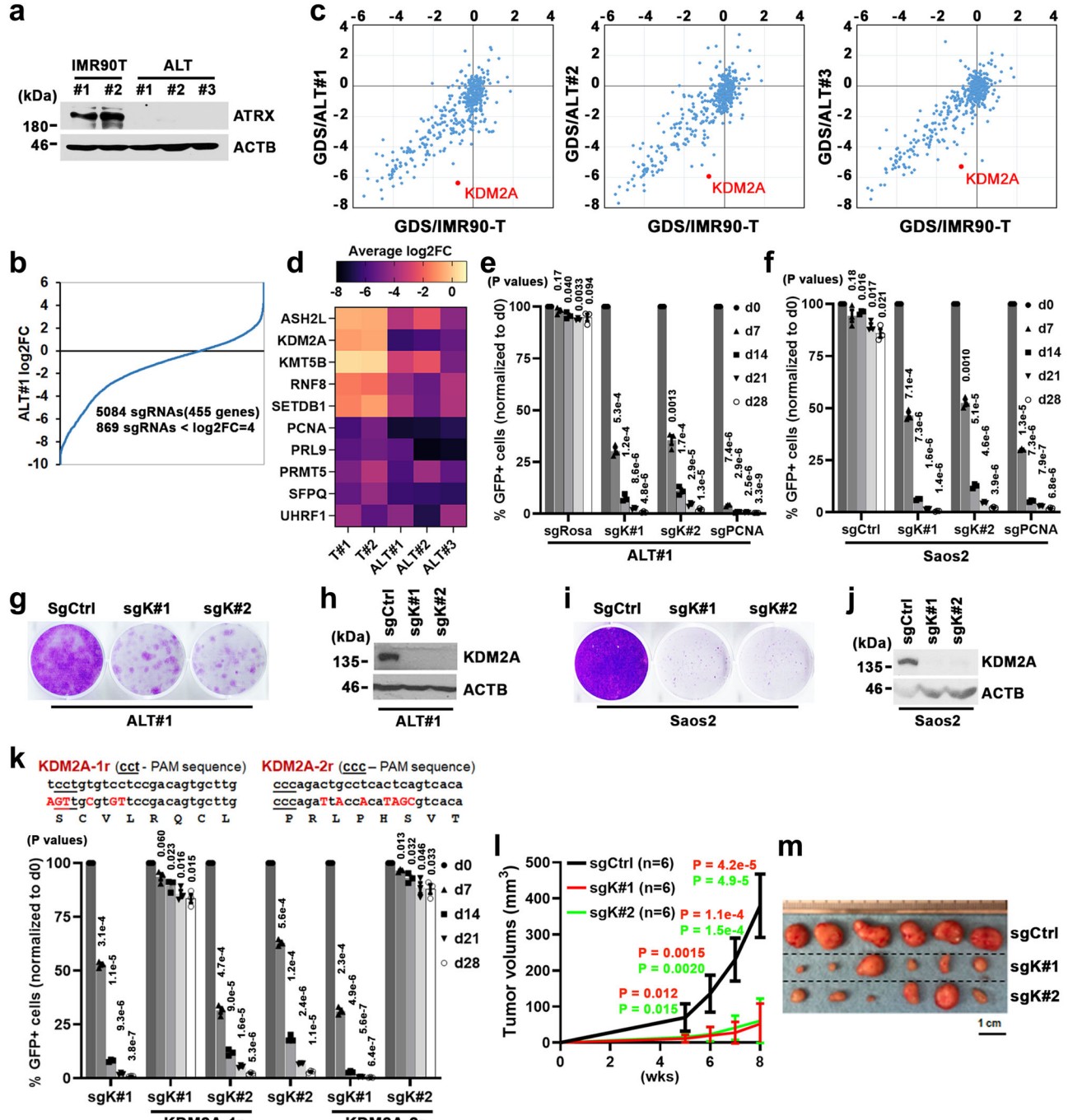

**Fig. 1 | CRISPR-based screens identify KDM2A as selectively essential in ALT-dependent cells. a** Western blot analysis of ATRX protein expression in whole-cell lysates prepared from the indicated cells. **b** Ranking of sgRNAs by log2 fold-change (log2FC) of abundance (ratio of start to endpoint) in ALT#1 cells. the x-axis shows targeting sgRNAs; the y-axis shows the $log_2FC$ of each targeting sgRNA after 16 population doublings. **c** Gene dependency scores (GDS) in IMR90-T (x-axis) versus ALT#1, ALT#2, or ALT#3 cells (y-axis). The GDS was calculated by averaging the log2FC of all sgRNAs targeting that gene. **d** Heatmap depicts log2FC of average sgRNA abundance of selected genes in indicated cells after 16 population doublings. **e, f** Competition-based proliferation assay of *KDM2A*-targeted sgK#1 and sgK#2 in ALT#1 (**e**) or Cas9-expressing Saos2 cells (**f**). A GFP reporter is linked to sgRNA expression. Plotted is the %GFP cells (normalized to the d0 measurement) at the indicated time points. The non-targeting sgCtrl was included as a negative and sgRNA targeting *PCNA* as a positive control. **g, h** Clonogenic assay (**g**) and KDM2A

western blot analysis (**h**) of sgCtrl, sgK#1, or sgK#2-transduced ALT#1 cells. Crystal violet staining was conducted on day 20 post-seeding. **i, j** Crystal violet-based clonogenic survival assay (**i**) and KDM2A western blot analysis (**j**) of sgCtrl, sgK#1, or sgK#2-transduced Saos2 cells. Crystal violet staining was conducted on day 24 post-seeding. **k** Competition-based proliferation assay of GFP-linked sgRNA in Saos2 cells complemented with either empty vector control or the CRISPR-resistant *KDM2A* synonymous mutants (*KDM2A-1r* or *KDM2A-2r*). Note, the *KDM2A-1r*-transduced cells are resistant to sgK#1 but sensitive to sgK#2 action; the *KDM2A-2r*-transduced cells are resistant to sgK#2 but sensitive to sgK#1 action. **l** Tumor growth curves of sgCtrl, sgK#1 or sgK#2-transduced Saos2 cells. Data were expressed as mean ± s.e.m. of six biological replicates; two-tailed unpaired *t*-test. **m** Image of tumors collected at week 8 post subcutaneous transplantation. In (**e, f, k**, data were expressed as mean ± s.e.m. of three independent experiments; two-tailed paired *t*-test. Source data are provided as a Source Data file.

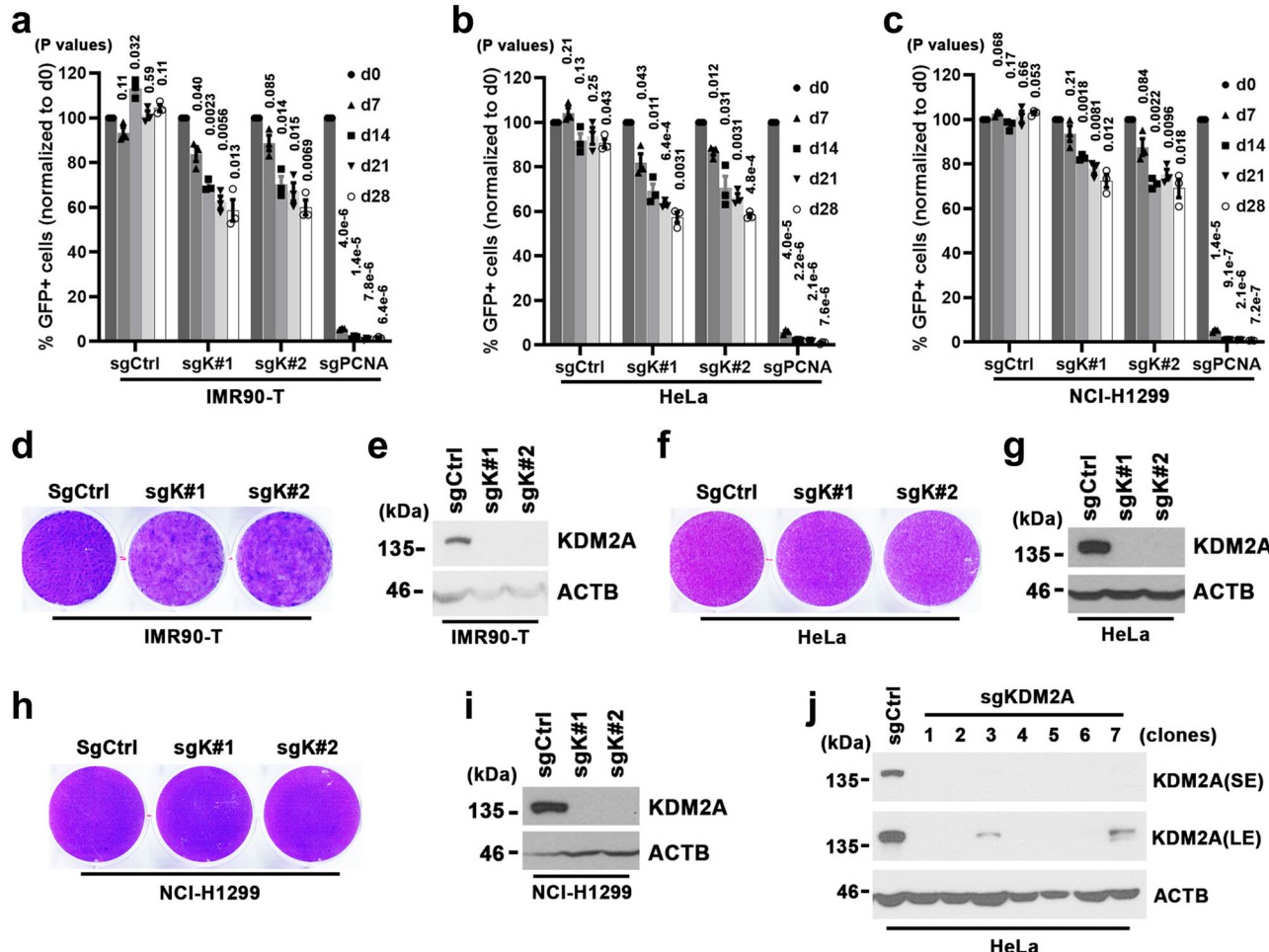

**Fig. 2 | KDM2A is dispensable for non-ALT cells. a–c** Competition-based proliferation assays of the indicated sgRNAs in IMR90-T (**a**), HeLa (**b**), or NCI-H1299 cells (**c**). A GFP reporter is linked to sgRNA expression. sgCtrl and sgPCNA were included as a negative or positive control, respectively. Data were expressed as mean ± s.e.m. of three independent experiments; two-tailed paired *t*-test. **d**, **e** Clonogenic assay (**d**) and KDM2A western blot analysis (**e**) of sgCtrl, sgK#1, or sgK#2-transduced IMR90-T cells. Crystal violet staining was conducted on day 22 post-seeding. **f**, **g** Clonogenic assay (**f**) and KDM2A western blot analysis (**g**) of sgCtrl, sgK#1, or sgK#2-transduced HeLa cells. Crystal violet staining was conducted on day 15 post-seeding. **h**, **i** Clonogenic assay (**h**) and KDM2A western blot analysis (**i**) of sgCtrl, sgK#1, or sgK#2-transduced NCI-H1299 cells. Crystal violet staining was conducted on day 15 post-seeding. **j** Western blot analysis of KDM2A and ACTB in whole-cell lysates prepared from indicated HeLa cell lines. The KDM2A-depleted lines (clone#1–7) were established from clonally isolated HeLa cells transduced with sgK#1. The sgCtrl-transduced HeLa line was included as a control. SE short exposure, LE long exposure. Source data are provided as a Source Data file.

NCI-H1299 of non-small cell lung carcinoma (Fig. 2c), MG63 of osteosarcoma (Supplementary Fig. 4a), MCF7 of breast cancer (Supplementary Fig. 4b), glioma cell lines A172, LN464, and U118 (Supplementary Fig. 4c–e), and primary lung fibroblast IMR90 cells (Supplementary Fig. 4f). As a positive control, sgRNAs targeting *PCNA* induced growth arrest in all the tested cell lines, ruling out the possibility of inefficient genome editing. These results were further verified by clonogenic assays of sgK#1 or #2-transduced IMR90-T (Fig. 2d, e), HeLa (Fig. 2f, g), or NCI-H1299 cells (Fig. 2h, i).

To determine whether KDM2A protein expression could be totally dispensable for non-ALT cell survival, we next applied the CRISPR/Cas9 system to deplete KDM2A in HeLa and LN464 cells. By western blot survey of clonally derived cultures, we identified 5 (out of 17) HeLa and 4 (out of 15) LN464 clones that were completely devoid of KDM2A protein expression (Fig. 2j and Supplementary Fig. 4g). These KDM2A-null cells were viable and proliferated at slightly slower rates than the parental HeLa or LN464 cells (Supplementary Fig. 4h, i), indicating that *KDM2A* is not a pan-essential gene. By contrast, we were not able to recover any KDM2A-null clones from sgK#1- or sgK#2-transduced ALT#1, Hs729, Saos2, or U2OS cell cultures. These findings support

KDM2A as a potential therapeutic target selectively for ALT-dependent human cancers.

## ALT cell growth depends on multiple functional activities of KDM2A

As a modular protein, KDM2A consists of a variety of structural motifs that serve different activities (Fig. 3a). To map the critical KDM2A protein domains for its ALT-supporting function, we synthesized a CRISPR exon-tilling library that comprised 492 sgRNAs targeting the entire *KDM2A* open reading frame (Supplementary Data 2). The tiling library was transduced into Saos2, ALT#1, ALT#2, and control IMR90-T cells. Using massively parallel sequencing, we calculated the depletion fold of each sgRNA over 16 population doublings. Among the sgRNAs that induced the most robust growth inhibition phenotypes in ALT-dependent Saos2, ALT#1, and ALT#2, were enriched for those that target the exons encoding the N-terminal demethylase domain, CXXC-type zinc finger (ZnF), or PHD domain of KDM2A (Fig. 3b, c and Supplementary Fig. 5a), indicating that DNA binding and demethylase activities of KDM2A are required for mediating its ALT-supporting function. By contrast, the sgRNAs that target the C-terminal domain,

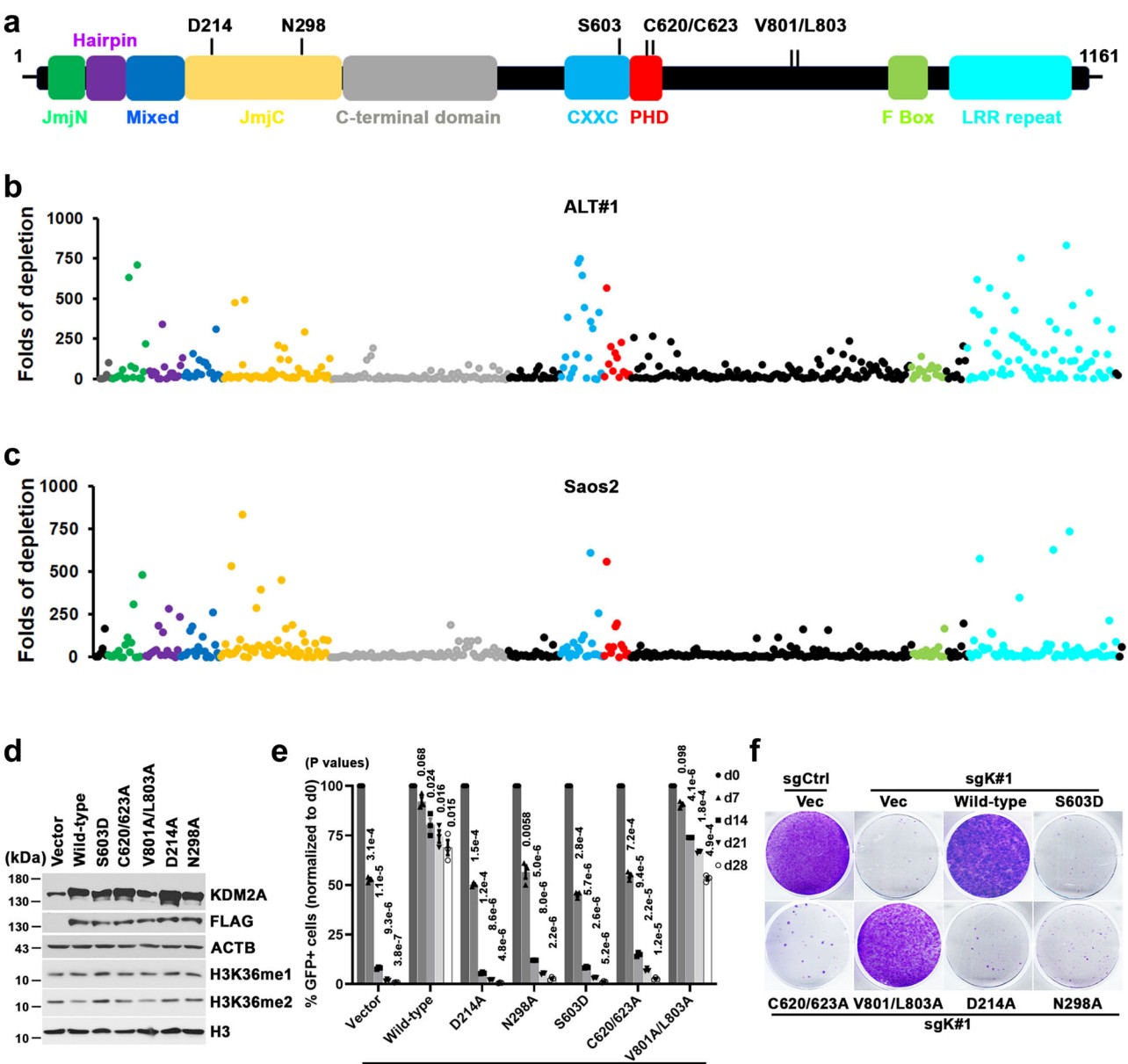

**Fig. 3 | DNA binding and demethylase activities of KDM2A are both required for its ALT-supporting function. a** Schematic of full-length KDM2A protein. The domains are labeled and color-coded. **b, c** CRISPR-based KDM2A tiling assay in ALT#1 (**b**) and Saos2 cells (**c**). Plotted is the fold changes of sgRNA abundance (ratio of start to endpoint) after 16 population doublings in culture. x-axis shows targeting sgRNAs and the domain location of each sgRNA within the KDM2A protein is indicated by colors; the y-axis shows the fold changes of each targeting sgRNA following the culture period. **d** Western blot analysis of KDM2A, Flag, and ACTB in whole-cell lysates prepared from Saos2 cells transduced with indicated constructs.

The Flag-tagged wild-type and KDM2A mutants were generated from the CRISPR-resistant *KDM2A-1r* construct. **e** Competition-based proliferation assay of *KDM2A*-targeted sgK#1 (linked with GFP expression) in Saos2 cells transduced with the indicated constructs. The bar graphs are expressed as mean ± s.e.m. of three independent experiments; two-tailed paired *t*-test. **f** Clonogenic assay of sgK#1 in Saos2 cells transduced with the indicated CRISPR-resistant wild-type or *KDM2A* mutants. Crystal violet staining was conducted on day 24 post-cell seeding. Source data are provided as a Source Data file.

F-Box, and other linker regions had less pronounced growth inhibition effects, suggesting that the E3 ligase activity of KDM2A is likely not required for sustaining ALT cell growth. Finally, none of the sgRNAs induced robust dropout phenotypes in the control IMR90-T cells (Supplementary Fig. 5b), further supporting that KDM2A is a selective vulnerability of ALT-dependent cells.

To validate the exon-tiling scan results, we next transduced Sao2 cells with sgK#1-resistant *KDM2A* cDNAs encoding wild-type or mutants defective of DNA binding (S603D)[30,31], PHD domain structural integrity (C620/623A)[30], HP1 protein interaction (V801A/L803A)[30], or demethylase activity (D214A or N298A)[25]. Despite being expressed at

comparable levels, western blot analysis found that only cells transduced with wild-type or the V801A/L803A mutant showed visible H3K36me2 reduction as compared to the vector transduced control cells (Fig. 3d). Consistently, competition and clonogenic-based proliferation assays revealed that complementation of wild-type or V801A/L803A, but not other mutants, rescued sgK#1-induced growth inhibition in Saos2 cells (Fig. 3e, f), indicating that HP1 protein interaction activity is dispensable for its ALT-supporting function.

The other identified KDM2A essential region in the exon-tilling scan of ALT cells was the C-terminal leucine-rich repeats (LRR). To test its ALT-supporting function, we constructed a KDM2A ΔLRR mutant

(aa 1-aa 945) that is deleted of the LRR motif. Consistent with the exon-tiling scan results, a competition-based proliferation assay in Saos2 cells found that complementation of the CRISPR-resistant ΔLRR mutant was not able to rescue the sgK#1-induced growth inhibition phenotype (Supplementary Fig. 6a, b).

## KDM2A physically binds to ALT telomeres

In human cancers and immortalized cell lines, ALT activation is closely associated with genetic alterations that affect the histone H3.3 cha-perone ATRX-DAXX complex[11,12,19,32]. Consistently, a western blot sur-vey of cell lines used in this study revealed that ATRX protein was broadly expressed in the non-ALT cells but absent in ALT-dependent cells (Fig. 4a). By contrast, KDM2A protein was expressed across the panel of cell lines regardless of their tissues of the origin or ATRX expression status, suggesting that their selective KDM2A dependency is not due to differential protein expression.

To explore whether KDM2A is a synthetic vulnerability of ATRX deficiency independently of its ALT status, we generated the *ATRX* knockout cells from TERT-transduced IMR90-T or LN464-T cells (Fig. 4b and Supplementary Fig. 7a). These clonally derived ATRX-null IMR90-T or LN464-T cells (referred to as dATRX#1 and #2) were viable and proliferated at slightly slower rates than their respective control cells (Supplementary Fig. 7b, c). Consistent with our previous findings[22], these TERT-overexpressed ATRX-null cells exhibited low levels of telomere dysfunction and TIF formation (Fig. 4c, d and Sup-plementary Fig. 7d, e). Competition-based proliferation assays further revealed that compared to ALT#1 cells, where KDM2A inhibition caused robust growth arrest, targeting these *ATRX*-knockout IMR90-T or LN464-T cells by *KDM2A*-specific sgRNAs only moderately affected their fitness (Fig. 4e and Supplementary Fig. 7f), suggesting that KDM2A is not a synthetic lethal vulnerability of simple ATRX deficiency.

Among the 21 histone lysine demethylases reported in the litera-ture, notably, our screen identified KDM2A as the only one essential for ALT cell growth (Supplementary Fig. 7g). To examine whether KDM2A may act physically at telomeres, we transduced Flag-tagged KDM2A into Saos2, ALT#1 and HeLa cells. Consistent with a previous study[33], telomere dot-blot analysis of anti-Flag chromatin immunoprecipita-tion (ChIP) revealed significant Flag-KDM2A enrichment at telomeres, preferentially in ALT cells (Fig. 4f-h). By comparison, anti-Flag ChIP/ Alu dot-blot analysis did not detect the ALT-preferential enrichment (Supplementary Fig. 8a, b). Finally, anti-H3K36me2 ChIP/telomere dot-blot analysis of KDM2A-depleted Saos2 and ALT#1 cells revealed sig-nificantly increased levels of telomere H3K36me2 as compared to the sgCtrl-transduced cells (Fig. 4i-k and Supplementary Fig. 8c-e). These results together support direct involvement of KDM2A in ALT-directed telomere maintenance.

## KDM2A facilitates the chromosomal segregation of ALT cells

ALT cells are characterized by persistent telomere DNA replication stress and rely on recombination-based DNA repair pathways to elongate their telomeres[5,8,23]. Cell cycle analysis of KDM2A-depleted ALT#1 or Saos2 cells found a markedly elevated G2/M phase accu-mulation as compared to their respective control cells (Fig. 5a and Supplementary Fig. 9a, b). Despite its ALT-supporting function, sur-prisingly, depletion of KDM2A did not seem to significantly affect many ALT-associated activities. For example, quantitation of control and KDM2A-depleted ALT#1 or Saos2 cells showed comparable levels of APB formation (Fig. 5b, c and Supplementary Fig. 9c, d), a hallmark of ALT activation[34]. Similarly, analysis of telomere length and C-rich extrachromosomal telomere repeats (C-circles) in control and KDM2A-depleted ALT#1 or Saos2 cells also did not reveal significant changes in telomere length heterogeneity and C-circle formation (Fig. 5d, e and Supplementary Fig. 9e, f).

To investigate whether KDM2A inactivation affects ALT-directed telomere DNA synthesis, sgCtrl, and sgK#1-transduced ALT#1 cells

were synchronized to the G2 phase with sequential thymidine and CDK1 inhibitor Ro-3306 treatment before 5-ethynyl-2´-deoxyuridine (EdU) labeling. Interestingly, the following assay of ALT telomere DNA synthesis in APBs (ATSA) found no significant difference in their levels of telomere EdU incorporation (Fig. 5f, g). A similar observation was also made in comparing sgCtrl and sgK#1-transduced Saos2 cells (Supplementary Fig. 9g, h), suggesting that KDM2A acts downstream of recombination-directed telomere DNA synthesis.

KDM2A depletion in ALT cells induces G2/M phase accumula-tion. This abnormal cell cycle distribution could be caused by cell cycle arrest or dysfunctional mitosis. To monitor mitotic cell division by time-lapse live cell imaging, the sgCtrl or sgK#1-transduced GFP-H2B-expressing ALT#1 cells were synchronized by sequential thy-midine and CDK1 inhibitor treatment before timed release. For the control sgRNA-transduced ALT#1 cells, a majority (102/134; 76%) that had entered mitosis during the imaging periods underwent normal chromosomal segregation and cytokinesis (Supplementary Movie 1). By comparison, of the 112 KDM2A-depleted cells that were tracked for their mitotic division, we found that 74% (83 of 112) of them displayed aberrant chromosomal segregation and eventually under-went mitotic catastrophe, as indicated by their hyper-condensed chromatin aggregates and/or DNA fragmentation (Fig. 5h, i and Supplementary Movie 2). These gross mitotic failures were also visualized in the live imaging of GFP-H2B-expressing ALT#2 cells following KDM2A inactivation (Supplementary Fig. 10a, b and Sup-plementary Movies 3, 4). By contrast, depletion of KDM2A in GFP-H2B-expressing IMR90-T cells did not significantly affect their mitotic division (Supplementary Fig. 10c, d and Supplementary Movies 5, 6).

Mitotic catastrophe represents a regulated mechanism that responds to aberrant mitoses by removing damaged cells from the cycling population[35,36]. Indeed, analysis of mitotic outcomes of the KDM2A-depleted ALT#1 cells revealed a significantly elevated mitotic death (Supplementary Fig. 10e, f). Western blot analysis further uncovered an increased level of apoptosis in sgK#1-transduced ALT#1 cells, as evidenced by the cleaved PARP1 production (Supple-mentary Fig. 10g). These findings suggest that KDM2A functions to facilitate mitotic chromosomal segregation of ALT cells.

## KDM2A is required for ALT multitelomere de-clustering

ALT-directed telomere synthesis occurs within APBs where recombi-nogenic telomeres from different chromosomes cluster together[8,37,38]. This process is cell cycle-regulated, and the clustered telomeres must be disassembled prior to anaphase to ensure proper chromosome segregation[38]. To test whether KDM2A is involved in the regulation of ALT telomere de-clustering, we synchronized the sgCtrl and sgK#1-transduced ALT#1 cells to the G2 phase. The immuno-FISH analysis of the G2-synchronized cells revealed comparable levels of APB forma-tion (Fig. 6a-c), suggesting that KDM2A loss did not affect telomere clustering and APB assembly. Following timed release from the CDK1 inhibitor block, we found that a majority of the control ALT#1 cells that entered mitosis and were marked by H3-Ser10 phosphorylation (pH3S10) had cleared their multitelomere clusters (Fig. 6a-c). By comparison, ~73% of mitotic ALT#1 cells that were depleted of KDM2A retained telomere cluster foci and eventually underwent aberrant segregation and mitotic catastrophe (Fig. 6d), indicating a defect in telomere de-clustering. Similar phenotypes were also observed in KDM2A-depleted Saos2 (Supplementary Fig. 11a-d), ALT#2 (Supple-mentary Fig. 11e-g), and G292 cells (Supplementary Fig. 11h-j).

To further validate KDM2A's role in the regulation of ALT telo-mere de-clustering, we conducted a reconstitution experiment. Indeed, the complementation of CRISPR-resistant *KDM2A* cDNAs into ALT#1 cells fully rescued the sgK#1-induced telomere segregation defects (Supplementary Fig. 11k-m). In addition, re-expression of ATRX in ALT#1 cells suppressed KDM2A depletion-induced

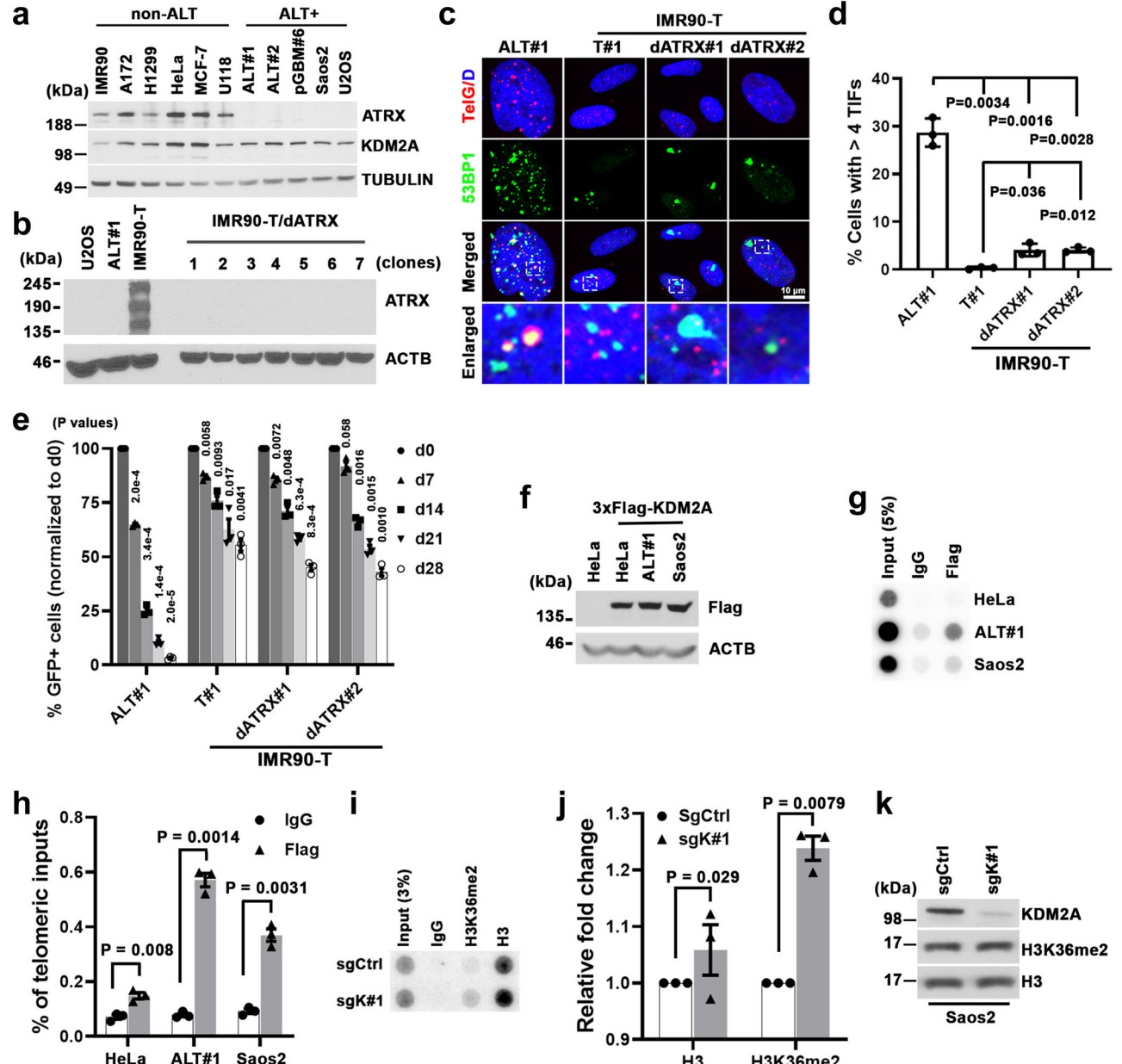

**Fig. 4 | KDM2A regulates H3K36me2 at ALT telomeres. a** Western blot analysis of ATRX protein expression in whole-cell lysates prepared from the indicated non-ALT and ALT-dependent (ALT+) cells. **b** Western blot analysis of ATRX expression in whole-cell lysates prepared from control or ATRX-depleted IMR90-T cells. The ATRX-depleted lines (dATRX clone#1–7) were established from clonally isolated IMR90-T cells transduced with ATRX-targeted sgRNA. The lysates from U2OS and ALT#1 cells were included as negative controls of ATRX expression.
**c** Representative immuno-FISH images of 53BP1 and telomeres in ALT#1, IMR90-T (T#1), or ATRX-depleted IMR90-T (dATRX#1 and dATRX#2) cells. Scale bar, 10 μm.
**d** Percentages of cells containing ≥4 53BP1-associated telomere dysfunction-induced foci (TIFs). Data were expressed as mean ± s.e.m. of three independent experiments; two-tailed unpaired *t*-test. **e** Competition-based proliferation assay of KDM2A-targeted sgK#1 (linked with GFP expression) in the indicated cell lines. Data were expressed as mean ± s.e.m. of three independent experiments. **f** Western blot analysis of Flag and ACTB (loading control) using whole-cell lysates prepared from

control or Flag-tagged wild-type KDM2A-transduced HeLa, ALT#1, or Saos2 cells. **g**, **h** Telomere dot-blot analysis (**g**) and quantification (**h**) of anti-Flag or IgG chromatin immunoprecipitation (ChIP) in the indicated cell lines. The IgG ChIP was included as a control for non-specific signals. The input and ChIP DNAs processed against the indicated antibodies were assayed by dot-blotting and hybridized with a $^{32}$P-labeled TelG probe. The relative enrichment was calculated after the normalization of ChIP DNA signals to the respective input DNA signals. Data were expressed as mean ± s.e.m. of three independent experiments; two-tailed paired *t*-test. **i**, **j** Telomere dot-blot analysis (**i**) and quantification (**j**) of anti-H3K36me2, anti-H3, or IgG ChIP in sgCtrl or sgK#1-transduced Saos2 cells. The relative enrichment was calculated after normalization to the respective ChIP DNA signals of sgCtrl-transduced samples. Data were expressed as mean ± s.e.m. of three independent experiments; two-tailed paired *t*-test. **k** Western blot analysis of KDM2A, H3K36me2, and H3 in cell lysates prepared from sgCtrl or sgK#1-transduced Saos2 cells. Source data are provided as a Source Data file.

multitelomere cluster formation and segregation dysfunction (Supplementary Fig. 12a–c), confirming that KDM2A is required for post-recombination ALT telomere de-clustering.

To characterize those aberrantly retained M-phase multitelomere clusters, we next analyzed their association with telomere-binding proteins. In line with the previous reports[38,39], mitotic telomeres of the control ALT#1 cells were largely condensed and exhibited reduced interaction with telomere-binding protein TRF1 (Fig. 6e, f). By contrast, the aberrantly clustered mitotic telomeres in the KDM2A-depleted ALT#1 cells were still strongly associated with TRF1, suggesting

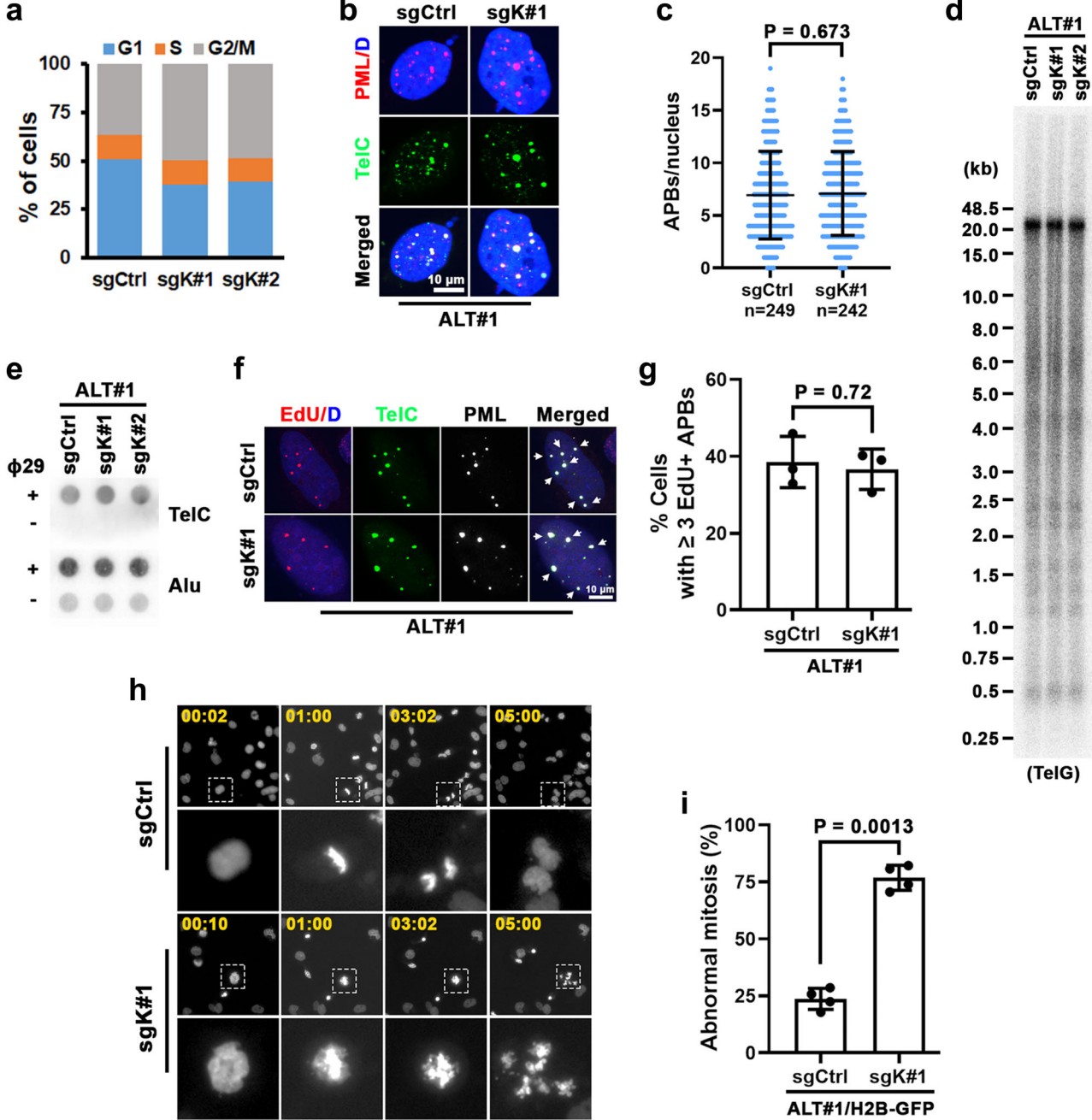

**Fig. 5 | KDM2A depletion disrupts ALT chromosomal segregation and mitotic division. a** Cell cycle distribution analysis of sgCtrl, sgK#1 or sgK#2-transduced ALT#1 cells. **b** Representative immuno-FISH images of ALT-associated PML bodies (APBs) in sgCtrl or sgK#1-transduced ALT#1 cells. Scale bar = 10 μm. **c** Percentages of cells containing ≥4 APBs are expressed as mean ± s.e.m. of three independent experiments; two-tailed unpaired *t*-test. **d** Telomere restriction fragment analysis of telomere length of sgCtrl, sgK#1, or sgK#2-transduced ALT#1 cells. Genomic DNAs prepared from the indicated cells were assayed by a [32]P-labeled TelG probe. **e** C-circle assays of sgCtrl, sgK#1 or sgK#2-transduced ALT#1 cells. Genomic DNAs were prepared from the indicated cells and assayed by a [32]P-labeled TelC probe. **f** Representative immuno-FISH images of EdU colocalized APBs (EdU-APBs) in sgCtrl or sgK#1-transduced ALT#1 cells. Cells were synchronized in G2 with

sequential thymidine and CDK1 inhibitor treatment and then labeled with EdU for 2 h. EdU was assayed by Click-It reaction, PML was analyzed by IF, and telomeres were detected by FISH. The arrows denote EdU-APB foci. **g** Percentages of cells containing ≥3 EdU-APB foci. Data were expressed as mean ± s.e.m. of three independent experiments; two-tailed unpaired *t*-test. **h** Representative frames of time-lapse fluorescence live cell imaging of GFP-H2B-expressing ALT#1 cells transduced with sgCtrl or sgK#1. The cells were synchronized in G2 with sequential thymidine and CDK1 inhibitor treatment before timed release into the M phase. Time in minutes is shown in the upper left corners. **i** Percentages of aberrant mitosis are expressed as mean ± s.e.m. of four independent experiments; two-tailed unpaired *t*-test. Source data are provided as a Source Data file.

compromised telomere condensation. Similar findings were also obtained in KDM2A-depleted Saos2 cells (Supplementary Fig. 13a, b).

Intermediates of homologous recombination are potential sources of chromosome missegregation if not removed before anaphase[40,41]. The Bloom's syndrome protein BLM is a RecQ family

helicase that drives ALT-associated telomere synthesis and intermediate telomere structure processing[41–44]. Consistent with our finding that KDM2A inactivation does not affect recombination-directed telomere DNA synthesis, immuno-FISH analysis of G2-synchronized control and KDM2A-depleted ALT#1 or Saos2 cells found comparable

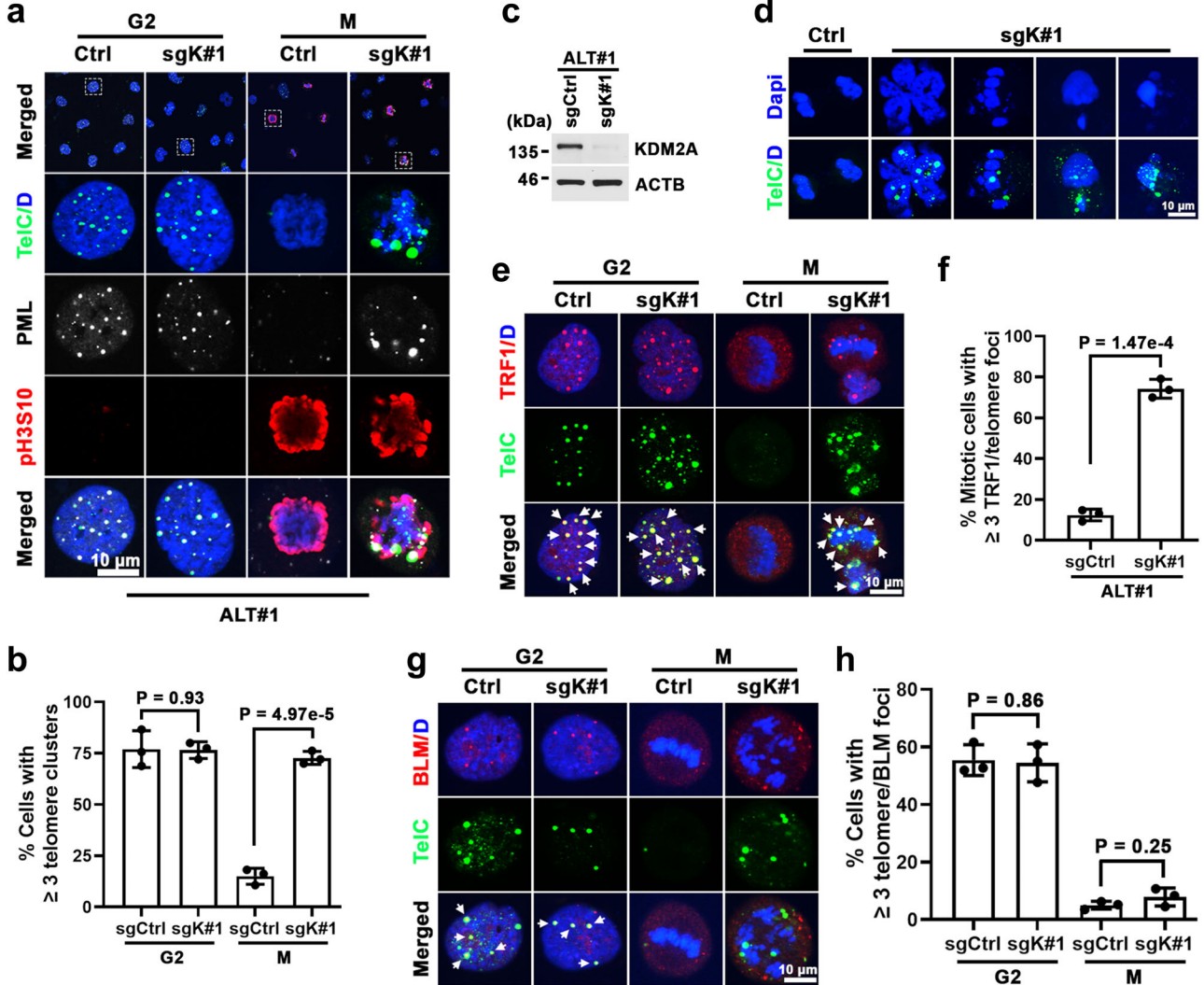

**Fig. 6 | KDM2A promotes ALT telomere de-clustering after recombination.**
**a** Representative immuno-FISH images of multitelomere clusters in G2 or mitotic
ALT#1 cells transduced with sgCtrl or sgK#1. The G2 cells were synchronized from
sequential thymidine and CDK1 inhibitor treatment. The mitotic (M) cells were from
G2-synchronized cells upon 1 h release from the CDK1 inhibitor. PML and mitotic
marker phospho-Histone H3-Ser10 (pH3S10) were analyzed by IF and telomeres
were detected by FISH. **b** Percentages of cells containing ≥3 multitelomere cluster
foci. **c** Western blot analysis of KDM2A and ACTB in whole-cell lysates prepared
from sgCtrl or sgK#1-transduced ALT#1 cells. **d** Representative images of abnormal
mitotic multitelomere clusters in sgK#1-transduced ALT#1 cells. G2-synchronized
ALT#1 cells transduced with sgCtrl or sgK#1 were released into mitosis for 2 h.
Telomeres were detected by FISH. **e** Representative immuno-FISH images of TRF1-
telomere association in G2 or M-phase ALT#1 cells transduced with sgCtrl or sgK#1.
The M-phase cells were from G2-synchronized cells upon 1 h release from the CDK1
inhibitor. TRF1 was analyzed by IF and telomeres were detected by FISH. The arrows
denote TRF1-associated telomere foci. **f** Percentages of mitotic cells containing ≥3
TRF1-associated telomere foci. **g** Immuno-FISH analysis of BLM and telomeres
colocalization in G2- or M-phase ALT#1 cells transduced with sgCtrl or sgK#1. BLM
was analyzed by IF and telomeres were detected by FISH. The arrows denote BLM-
associated telomere foci. **h** Percentages of cells containing ≥3 BLM-associated tel-
omere foci. Note, all bar graphs are expressed as mean ± s.e.m. of three indepen-
dent experiments; two-tailed unpaired t-test. Scale bar, 10 µm. Source data are
provided as a Source Data file.

levels of telomere-associated BLM foci formation (Fig. 6g, h and Sup-
plementary Fig. 13c, d). Upon mitotic entry following release from the
CDK1 inhibitor block, as expected, telomeres in the sgCtrl-transduced
control cells was dissociated from BLM binding. Similarly, BLM was
also cleared from the aberrant telomere clusters of mitotic ALT#1 or
Saos2 cells that were depleted of KDM2A, indicating that those
abnormal telomere clusters are not unresolved DNA recombination
intermediates.

### KDM2A promotes SENP6-mediated ALT telomere de-SUMOylation
SMC5/6 complex-activated protein SUMOylation is critical for ALT-
directed telomere clustering and APB formation[45]. Indeed, we found
that SMC5 colocalized with APBs and its knock-down in ALT#1 cells

blocked APB and telomere cluster foci formation (Supplementary
Fig. 14a, b). Moreover, compared to the scrambled control siRNA
treatment, transduction of SMC5 siRNA in sgK#1-transduced
ALT#1 cells strongly attenuated the aberrant M-phased telomere
clusters following KDM2A depletion (Fig. 7a–c), suggesting that
KDM2A functions downstream of SMC5/6 action.

ALT telomere clustering and phase-separated APB assembly rely
on SUMOylation of telomere-associated proteins[41,44,45]. Notably,
immuno-FISH analysis of G2-synchronized control and KDM2A-
depleted ALT#1 cells revealed comparable levels of SUMO2/3-asso-
ciated telomere foci (SUMO-T) formation (Fig. 7d, e), suggesting that
KDM2A is not crucial for telomere protein SUMOylation. Upon release
from the CDK1 inhibitor block, the control ALT#1 cells that had entered
mitosis largely cleared their SUMO-T foci and multitelomere clusters.

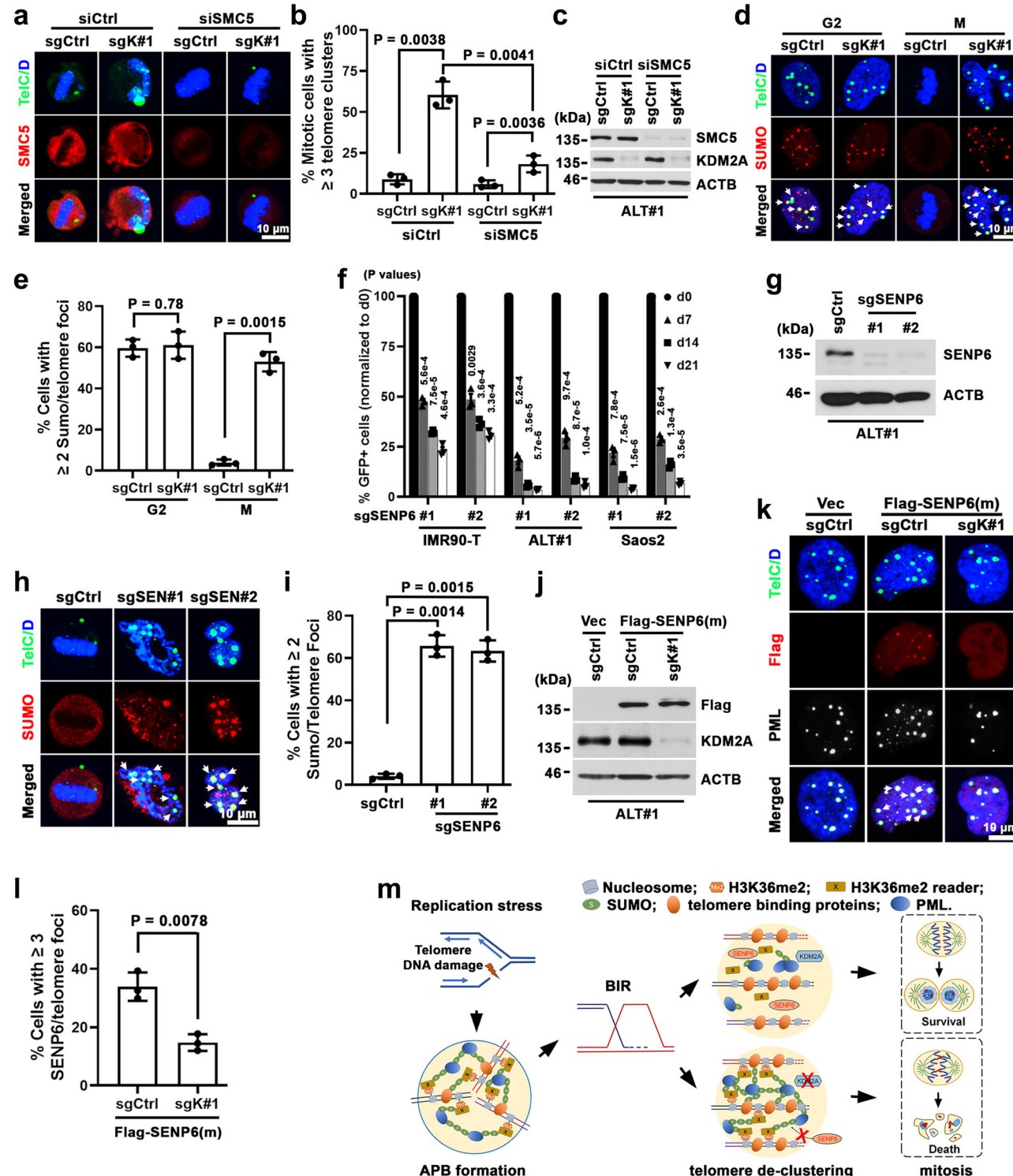

By contrast, a large portion of sgK#1-transduced mitotic ALT#1 cells maintained their SUMOylation at the aberrantly retained multi-telomere foci (Fig. 7d, e), suggesting a role of KDM2A in the regulation of telomere de-SUMOylation. Similar results were also obtained in KDM2A-depleted Saos2 cells (Supplementary Fig. 14c, d).

SUMO deconjugation from substrates is catalyzed by a group of SENP family of isopeptidases[46]. To interrogate the role of de-SUMOylation in ALT-directed telomere maintenance, we assembled a focused sgRNA sub-library targeting the seven human SENP family members (10 sgRNAs/gene) (Supplementary Table 1). The dropout screens of the isogenic ALT (ALT#1, #2, and #3) and paired IMR90-T

control cells (IMR90-T#1 and #2) scored SENP6 as the only member that was differentially required by ALT cells (Supplementary Fig. 14e). Competition-based proliferation assays of two independent SENP6-sgRNAs (sgSEN#1 and #2) confirmed its essentiality to ALT-dependent ALT#1 and Saos2 cells (Fig. 7f). Notably, depletion of SENP6 also significantly affected the growth of IMR90-T#1 cells, although to a lesser extent than ALT#1 and Saos2 cells. To examine whether SENP6 loss interferes with the de-SUMOylation at ALT telomeres, the sgCtrl, sgSEN#1, or sgSEN#2-transduced ALT#1 cells were synchronized to the G2 phase by the sequential thymidine and CDK1 inhibitor treatment. Immuno-FISH analysis of the G2-synchronized control and SENP6-

**Fig. 7 | KDM2A facilitates SENP6-mediated ALT telomere de-SUMOylation.**
**a** Representative immuno-FISH images of multitelomere clusters in control or
SMC5 siRNA transfected mitotic ALT#1 cells expressing sgCtrl or sgK#1. The mitotic
(M) cells were prepared from G2-synchronized cells after 1 h release from the CDK1
inhibitor. The arrows denote abnormal mitotic telomere cluster foci. **b** Percentages
of mitotic cells containing ≥3 telomere cluster foci are expressed as mean ± s.e.m.
of three independent experiments; two-tailed unpaired *t*-test. **c** Western blot ana-
lysis of SMC5 expression in whole-cell lysates prepared from the siCtrl or siSMC5-
transfected ALT#1 cells. **d** Representative immuno-FISH images of telomere
SUMOylation in sgCtrl or sgK#1-transduced ALT#1 cells. The mitotic (M) cells were
from G2-synchronized cells upon 1 h release from the CDK1 inhibitor. The arrows
denote SUMO2/3-associated telomere foci. **e** Percentages of cells containing ≥2
SUMO-associated telomere foci. Data were expressed as mean ± s.e.m. of three
independent experiments; two-tailed unpaired *t*-test. **f** Competition-based pro-
liferation assay of the indicated *SENP6*-targeted sgRNAs (sgSEN#1 and sgSEN#2) on
the fitness of IMR90-T, ALT#1 and Saos2 cells. The data are expressed as mean ±

s.e.m. of three independent experiments; two-tailed paired *t*-test. **g** Western blot
analysis of SENP6 in whole-cell lysates prepared from sgCtrl, sgSEN#1 or sgSEN#2-
transduced ALT#1 cells. **h** Immuno-FISH analysis of telomere SUMOylation in
mitotic ALT#1 cells transduced with sgCtrl, sgSEN#1, or sgSEN#2. The mitotic cells
were from G2-synchronized cells upon 1 h release from the CDK1 inhibitor. The
arrows denote SUMO2/3-associated telomere foci. **i** Percentages of mitotic cells
containing ≥2 SUMO-associated telomere foci are expressed as mean ± s.e.m. of
three independent experiments; two-tailed unpaired *t*-test. **j** Western blot analysis
of Flag and ACTB (loading control) in whole-cell lysates prepared from the indi-
cated ALT#1 cells transduced with Flag-tagged SENP6[C1030A] mutant. **k** Immuno-FISH
analysis of telomere and SENP6[C1030A] association in Flag-SENP6[C1030A]-expressing
ALT#1 cells transduced with sgCtrl or sgK#1. The arrows denote SENP6[C1030A]-asso-
ciated telomere foci. **l** Percentages of cells containing ≥3 Flag-SENP6[C1030A]-asso-
ciated telomere foci are expressed as mean ± s.e.m. of three independent
experiments; two-tailed unpaired *t*-test. **m** A model illustrating the key events in the
ALT pathway. Source data are provided as a Source Data file.

depleted ALT#1 cells found comparable levels of SUMO-T foci forma-
tion (Supplementary Fig. 14f, g), suggesting that SENP6 is not crucial
for ALT telomere SUMOylation or APB formation. After being released
from CDK1 inhibitor treatment, the control ALT#1 cells that
had entered mitosis largely cleared their telomere SUMOylation and
SUMO-T foci. By contrast, a large percentage of SENP6-depleted
mitotic ALT#1 cells retained SUMOylated multitelomere clusters
(Fig. 7g–i), reminiscent of KDM2A depletion. Similar phenotypes were
also observed in Saos2 cells that were depleted of SENP6 expression
(Supplementary Fig. 14h–l). These findings indicate that SENP6-
mediated de-SUMOylation is required for ALT telomere de-clustering
following recombination-directed telomere synthesis.

Since the inactivation of KDM2A and SENP6 in ALT cells induced a
similar telomere de-clustering phenotype, we next examined whether
KDM2A loss might interfere with ALT telomere protein de-
SUMOylation through blocking SENP6 recruitment to its substrates.
Notably, while SENP6 binds to its substrates transiently, mutation of
the catalytic cysteine to alanine (C1030A) stabilizes the interaction[47].
To test whether SENP6 physically interacts with ALT telomeres, we
expressed the *SENP6[C1030A]* mutant transgene in ALT#1 cells (Fig. 7j).
Analysis of the G2-synchronized ALT#1 cells found that the Flag-tagged
SENP6[C1030A] was indeed localized to the clustered ALT telomeres
(Fig. 7k). Importantly, quantitation of the telomere-associated
SENP6[C1030A] foci revealed that KDM2A depletion significantly dimin-
ished the telomere recruitment of SENP6[C1030A] (Fig. 7l). Similar results
were also obtained in Saos2 cells depleted of KDM2A expression
(Supplementary Fig. 14m–o), suggesting that KDM2A may function to
promote ALT telomere de-clustering by facilitating SENP6-mediated
telomere de-SUMOylation. Consistently, complementation of CRISPR-
resistant cDNAs encoding wild-type, but not KDM2A mutant defective
of demethylase (D214A) or chromatin binding (S603D), restored
SENP6[C1030A] telomere recruitment in sgK#1-transduced ALT#1 cells
(Supplementary Fig. 15a–c). Finally, it is worth noting that our co-
immunoprecipitation analysis uncovered no evidence of physical
interaction between KDM2A and SENP6 (Supplementary Fig. 16a).
Depletion of SENP6 in ALT#1 cells also did not affect KDM2A protein
SUMOylation as compared to the control ALT#1 cells (Supplementary
Fig. 16b). These findings suggest that KDM2A may indirectly regulate
SENP6 recruitment to ALT telomeres through modulating telomere
H3K36 methylation.

## Discussion

The implementation of the genetic concept of synthetic lethal or
synthetic lethal-like interaction holds great promise in anticancer tar-
get discovery. By undertaking chromatin regulator-focused genetic
screens in a well-controlled isogenic ALT cellular model system, we
identified histone lysine demethylase KDM2A as a selective molecular
vulnerability of ALT-dependent cells. We further demonstrate that

KDM2A is required for post-recombination ALT multitelomere de-
clustering. We show that depletion of KDM2A in ALT cells impairs
isopeptidase SENP6-mediated SUMO deconjugation at ALT telomeres.
Inactivation of KDM2A or SENP6 compromises post-recombination
ALT multitelomere de-SUMOylation and de-clustering that subse-
quently lead to mitotic chromosome missegregation and cell death.
Results from this study thus support efforts to develop KDM2A inhi-
bitors targeting ALT-dependent cancers.

Synthetic lethality provides a framework for targeting the loss of
function of tumor suppressor and DNA repair genes, as exemplified by
the success of PARP inhibitors in the treatment of BRCA1/2-deficient
tumors[48,49]. The advance of CRISPR-based screen methodology has
further enabled large-scale studies to profile synthetic lethal or syn-
thetic lethal-like interactions in a diverse collection of human cancer
cells across many genetic contexts[50–52]. But despite the fact that ALT is
utilized in a substantial fraction (5–10%) of human cancers, ALT cancer
cells were rarely included in those screening initiatives. The poor
representation is likely due to the paucity of ALT cancer cell lines
suitable for large-scale functional genomic screens. To overcome this
hurdle, we employed an in vitro ALT-immortalized isogenic model
system. The inclusion of isogenic telomerase-positive and ALT-positive
cells derived from the same genetic background greatly reduced
potential confounding variables such as cell type differences and co-
occurring genetic changes, making it easy to infer the genetic inter-
actions from a small panel of paired cell lines. Our study thus
demonstrates the applicability of well-controlled isogenic models in
identifying and prioritizing targets from genetic interaction screens.

Ideal anticancer therapies should demonstrate a robust ther-
apeutic window toward tumor cells by sparing normal cells and lim-
iting off-tumor toxicity. This study identifies KDM2A as a promising
therapeutic target that is selective for ALT-dependent cancers. Inacti-
vation of KDM2A in ALT cells causes robust cell growth inhibition by
inducing mitotic chromosome missegregation and cell death. By
comparison, our competition-based proliferation and total knock-out
assays of a panel of non-ALT cells of diverse tissue origins found that
KDM2A is largely dispensable for their growth and survival. Con-
sistently, conditional *Kdm2a* deletions in the mouse myeloid com-
partment revealed no discernible impact on the development and cell
maturation[53]. These findings suggest a potentially large therapeutic
window for future KDM2A-targeted therapies.

Emerging evidence suggests that ALT-directed telomere elonga-
tion emanates from telomere replication stress and proceeds through
a conservative BIR-like pathway[5,6,20,22,41]. However, the molecular sig-
nals that control ALT pathway initiation and termination still remain
largely unclear. For example, although APB formation is known to
depend on protein SUMOylation[45], how telomere replication stress
signals proceed to promote APB formation in ALT cells is poorly
understood. Equally unclear are the molecular events that govern the

multitelomere cluster dissolution after the ALT-directed telomere DNA synthesis. In this study, we found that post-recombination ALT telomere de-clustering requires SENP6-mediated de-SUMOylation. Our data further indicate that this process is regulated by KDM2A-directed telomere H3K36me2 demethylation, underscoring its importance in ALT-directed telomere maintenance.

Histone methylation pathways play important roles in orchestrating DNA damage signaling[54–56]. Among the total of 21 histone lysine demethylases, our genetic screen identifies KDM2A as the only one that is selectively essential for ALT cell growth. Consistent with a previous proteomic study[33], our telomere dot-blot assay of chromatin immunoprecipitation demonstrate the physical interaction of KDM2A with ALT telomeres, supporting a direct KDM2A involvement in ALT-directed telomere maintenance. KDM2A is a lysine-specific demethylase that targets lower methylation states of H3K36 (Kme1 and Kme2)[25,26]. Indeed, our data indicate that depletion of KDM2A in ALT cells leads to increased telomere H3K36me2. Interestingly, KDM2A is not required for ALT-directed telomere DNA synthesis. Instead, we found that KDM2A-mediated demethylation promotes post-recombination ALT telomere de-clustering, at least partly through facilitating SENP6-mediated telomere de-SUMOylation. Given that H3K36 methylation is a major chromatin change following DNA double-strand break (DSB)[57,58], we propose a model in which KDM2A functions as an epigenetic eraser of H3K36me2-dependent signaling that safeguards proper chromosome segregation by facilitating SENP6-mediated ALT telomere de-SUMOylation and de-clustering (Fig. 7m). Loss of KDM2A impairs SENP6 recruitment to ALT telomeres and thus their de-clustering, leading to mitotic chromosome missegregation. But despite that our data support a direct KDM2A involvement in ALT-directed telomere maintenance, we cannot exclude the possibility that KDM2A may contribute indirectly through transcriptional regulation of genes involved in the process. Also, noticeably, our co-immunoprecipitation analysis did not find evidence of physical KDM2A and SENP6 interaction. Future studies are needed to sort out the detailed molecular events that follow KDM2A action.

Our study nominates KDM2A as a selective molecular vulnerability of ALT-dependent cells. In human cancers, ALT activation is strongly associated with mutations of the chromatin modulator genes *ATRX* and *DAXX*[9–13]. But notably, our study reveals that KDM2A is not a synthetic lethal vulnerability with simple ATRX or DAXX loss. Instead, we found that KDM2A is critical for ALT-directed telomere maintenance, which is utilized by cells deficient in ATRX or DAXX. These findings suggest that KDM2A-mediated demethylation may play an important role in resolving the telomere replication stress and repair-associated intermediate structures following recombination-directed telomere synthesis. As replication stress-induced DNA damage and repair are common features of human cancers, it will be interesting in the future to evaluate the molecular functions of KDM2A and, more broadly, H3K36 methylation and demethylation in those processes.

## Methods

### Cell lines and plasmids
The human cell lines A172, G292, HEK293T, HeLa, Hs792, IMR90, MCF7, MG63, NCI-H1299, Saos2, U118, and U2OS were obtained from ATCC. The IMR90-T, ALT#1, #2, and #3 cells were derived from large T-transformed IMR90 cells as previously described in ref. [22]. The patient-derived *ATRX*-mutant glioblastoma cell line pGBM6 was established from collected tumor specimens after obtaining written informed consent preoperatively and approved by the Institutional Reviewer Boards of the Southwest Hospital (KY2020147). The human glioblastoma cell line LN464 was kindly provided by F. Furnari (University of California, San Diego). The clonally derived *KDM2A*-knockout HeLa and LN464 cell lines, or *ATRX*-depleted IMR90-T and LN464-T cells were generated using lentiviruses produced in HEK293T cells with lentiCRISPR-v2 vectors containing *KDM2A* or *ATRX* targeting sgRNA

and selected with blasticidin as previously described[22]. For the cDNA expression experiments, full-length cDNAs was cloned into a lentiviral expression vector pLU-IRES-Puro, -Blast, or -Neo vector containing 3xN terminal Flag or C-terminal GFP tag. CRISPR sgRNA-resistant synonymous or functional domain point mutations were introduced by PCR mutagenesis using NEBuilder HiFi DNA Assembly Master Mix (NEB, E2631). Stable cell lines were generated using the pLU vectors and selected with puromycin or blasticidin. Cell lines obtained were not authenticated and were tested negative for mycoplasma. In this study, all cells were cultured with respective media in a humidified 37 °C, 5% $CO_2$ incubator. All the sgRNAs targeting human genes were cloned into lentiCRISPR-v2 (Addgene, #52961), lentiCas9-Blast (Addgene, #52962), LRG2.1 (U6-sgRNA-GFP, Addgene, #108098), or LRPuro (U6-sgRNA-Puromycin) as indicated. Single sgRNAs were cloned by annealing two DNA oligos and ligating into a BsmB1-digested vector. To improve U6 promoter transcription efficiency, an additional 5' G nucleotide was added to all sgRNA oligo designs that did not already start with a 5' G. A list of sgRNA information is provided in Supplementary Table 2.

### Animal experiments
NSG (NOD.Cg-Prkdcscidll2rgtm1Wjl/SzJ) mice were purchased from Jackson laboratories. Mice were group-housed (up to 5 per cage) in individually ventilated cages with ad libitum access to food and acidified water (pH 2.5 to 2.8) in a temperature (22.2 ± 0.5 °C) and humidity (30–70%) controlled facility with 12/12-h light/dark cycle. The animal care and use program is accredited by AAALAC. All animal experiments were approved by the Weill Cornell Institutional Animal Care and Use Committee. For all mouse studies, mice of either sex were used, and mice were randomly allocated to experimental groups, but blinding was not performed. Mice (aged 5–8 weeks) were age-matched for tumor inoculation. Group sizes were selected on the basis of prior knowledge. For subcutaneous grafting, sgCtrl, sgK#1 or #2-transduced Saos2 cells were resuspended in 50% Matrigel (BD Bioscience, #356231) in PBS and ~5,000,000 cells were injected into each flank of NSG mice. Tumor growth was monitored and measured every 7 days by caliper, and volume was calculated by the formula: $V = \frac{4\pi}{3} * \frac{a}{2} * \frac{b}{2} * \frac{c}{2}$ (V tumor volume; a tumor length; b tumor width; c tumor height). The endpoints were determined on the basis of the level of animal discomfort and tumor sizes. The maximum allowable tumor size is 20 mm in diameter.

### Construction of pooled sgRNA library
A gene list of 455 chromatin modification-associated factors in the human genome was manually curated. 4–14 sgRNA were designed against the functional domains of each gene based on the domain sequence information retrieved from NCBI Conserved Domains Database. The design principle of sgRNA was based on previous reports and the sgRNAs with the predicted high off-target effect were excluded[59–61]. All of the sgRNAs oligos, including positive and negative control sgRNAs, were synthesized in a pooled format (CustomArray Inc) and PCR amplified. The library was constructed by cloning the PCR-amplified products into the BsmB1-digested LRPuro vector. The identity and relative representation of sgRNAs in the pooled plasmids were verified by a deep-sequencing-based analysis.

### Construction of CRISPR-based KDM2A exon-tiling library
A list of total of 492 sgRNAs that covered the entire open reading frame of *KDM2A* was designed by excluding the ones with predicted high off-target effect[59–61]. All of the sgRNAs oligos, including positive and negative control sgRNAs were synthesized in a pooled format (CustomArray Inc). The library was generated by cloning the PCR-amplified products of the synthesized sgRNA oligos into the BsmB1-digested LRPuro vector. The identity and relative representation of sgRNAs in the pooled plasmids were verified by a deep-sequencing-based analysis.

## Lentiviral transduction

Lentiviruses were produced by co-transfection of indicated plasmids and packaging vectors into HEK293T packaging cells, as previously described in ref. [22]. In brief, to generate lentivirus, $8 \times 10^6$ 293T cells in 100 mm tissue culture dishes were transfected with a mixture of 8.5 µg of plasmid DNA, 4 µg of pMD2.G, and 6 µg of psPAX2 packaging vectors, and 45 µl of 1 mg/mL Polyethylenimine (PEI 25000). The media was replaced 6–8 h post-transfection. The virus-containing supernatant were collected at 48 and 72 h post-transfection and pooled. For infection, virus-containing supernatant was mixed with the indicated cell lines supplied with 4 mg/mL polybrene and then centrifuged at 2000 rpm for 30 min at room temperature. Fresh media was changed 24 h post-infection. Antibiotics (10 µg/mL blasticidin, 2 µg/mL puromycin, 500 µg/mL G418, and/or 200 µg/mL hygromycin) were added 48 h post-infection when the selection was required.

## Pooled CRISPR-Cas9 and KDM2A exon-tiling screen

The pooled CRISPR-based negative selection and KDM2A exon-tiling screens were carried out as previously described with some modifications[62]. In brief, Cas9-expressing cells were infected with the lentiviral chromatin modification-associated factor- or KDM2A exon-tiling sgRNA library at an MOI 0.3–0.4 such that every sgRNA is represented in ~1000 cells. Fresh media was changed 24 h post-infection. At 48 h post-infection, puromycin (2 µg/mL) was added, and the infected cells were selected for 48–72 h. To maintain the representation of sgRNAs, the number of infected cells was kept at least 1000 times the sgRNA number in the library during the screen. Start-point cells were harvested at day 5 post-infection and served as a reference representation of the pooled sgRNA library. Cells were cultured for 16 population doublings and harvested as the end time point. Genomic DNA was extracted from cell pellets using QIAamp DNA Blood Maxi Kit (QIAGEN, #51194). The sgRNA cassette was PCR amplified from genomic DNA using Phusion High-Fidelity PCR Master Mix (New England Biolabs, M0531S). The amplified products were pooled and amplified again via PCR using primers harboring Illumina TruSeq adapters with i5 and i7 barcodes, and the resulting libraries were sequenced on an Illumina Nextseq500. The sequencing data was de-multiplexed. The read count of each sgRNA was calculated with no mismatches by comparing it to the reference sequence. Individual sgRNAs with a read count lower than 50 in the initial time point were discarded.

## Competition-based proliferation assay

The flow cytometry analysis (FACS)-based dropout assay was performed as previously described in ref. [22]. In brief, the indicated Cas9-expressing cell lines were transduced with a lentiviral LRG2.1 (U6-sgRNA-GFP, Addgene, #108098) sgRNA vector that co-expresses a GFP reporter. The percentage of sgRNA-transduced GFP-positive cell population in culture was measured at indicated time points using an LSRII flow cytometer (BD Biosciences). The change in GFP percentage was used to assess the proliferation of sgRNA-transduced cells relative to the non-transduced cells in the culture.

## Clonogenic-based proliferation assays

To evaluate the growth rate difference, indicated control and experimental cells were plated at 10,000–20,000 cells per well of a six-well plate. Cells were cultured with respective media in a humidified 37 °C, 5% $CO_2$ incubator for 15–25 days before being fixed with 10% formalin and stained with 0.1% crystal violet.

## Immuno-FISH assay

Indirect immunofluorescence (IF) combined with fluorescence in situ hybridization (FISH) analysis was performed as previously described in ref. [22]. Briefly, cells grown on coverslips were fixed for 15 min in 4% paraformaldehyde at RT, followed by permeabilization for 10 min in PBS with 0.3% Triton X-100 at RT. After washing with 1x PBS, cells were incubated for 60 min in blocking solution (1% BSA, 10% FBS, 0.2% fish gelatin, 0.1% Triton X-100, and 1 mM EDTA in 1x PBS) before immunostaining. Primary antibodies were prepared in blocking solution as following dilutions: 53BP1 (1:500, IHC-00001; Bethyl Laboratories), 53BP1 (1:300, AF1877; R&D Systems), BLM (1:250, Cat# A300-110A; Bethyl Laboratories), Flag (1:200, F1804; Sigma), γH2AX (1:2,000, A300-081A; Bethyl Laboratories), phospho-Histone H3 (Ser10) (1:200, #9701; Cell Signaling), PML (1:200, sc-966; Santa Cruz Biotechnology), PML (1:500, ab96051; Abcam), SMC5 (1:400, A300-236A; Bethyl Laboratories), SUMO2/3 (1:250, ab3742; Abcam), and TRF1 (1:200, ab10579; Abcam). After incubated with indicated primary antibodies at RT for 2 h, cells were washed three times with PBST (1x PBS containing 0.1% Tween-20) before being incubated with indicated secondary antibodies conjugated to fluorophores diluted in the same solution for 45 min at RT. After three washes with PBST, cells were fixed again with PFA for 10 min, then washed in 1x PBS, dehydrated in ethanol series (70, 95, 100%), and air-dried. Coverslips were denatured for 10 min at 85 °C in a hybridization mix [70% formamide, 10 mM Tris-HCl, pH 7.2, and 0.5% blocking solution (Roche)] containing 100 nM telomeric PNA probe TelC-FITC (F1009, PNA Bio), and hybridization was continued at RT for 2 h in the dark moisturized chambers. Coverslips were washed three times with Wash solution (70% formamide, 2x SSC) for 10 min each and in 2x SSC, 0.1% tween-20 three times for 5 min each. During the second wash, cells were stained with DAPI. Slides were mounted with VectorShield (Vector Laboratories). Images were captured with a 60x lens on an Olympus FLUOVIEW laser scanning confocal microscope (Olympus).

## Protein extraction and western blotting

Cells were lysed in RIPA buffer (150 mM NaCl, 50 mM Tris, 0.5% Na-Deoxycholate, 0.1% SDS, and 1% NP-40), and an equal amount of protein was resolved using Nupage Novex 4–12% Bis-Tris Gel (Thermo Fisher Scientific) as previously described in ref. [22]. Primary antibodies used were β-ACTIN (ACTB) (1:5,000, #2228; Sigma), ATRX (1:1,000, sc-15408; Santa Cruz Biotechnology), Flag (1:1,000, F1804; Sigma), H3 (1:2,000, ab1791; Abcam), H3K36me2 (1:1,000, 2901; Cell Signaling), H3K36me3 (1:2,000, 61101; Active Motif), KDM2A (1:1,000, A301-476A; Bethyl Laboratories), SENP6 (1:500, HPA024376; Sigma-Aldrich), SMC5 (1:2000, A300-236A; Bethyl Laboratories), and TUBULIN (1:2,000, ab15246; Abcam). Secondary antibodies used were donkey anti-rabbit HRP (1:1,000, sc-2077; Santa Cruz Biotechnology), donkey anti-mouse HRP (1:1,000; sc-2096, Santa Cruz Biotechnology), and donkey anti-goat HRP (1:1,000, sc-2056, Santa Cruz Biotechnology). Antibody signal was detected using the ECL Western Blotting Substrate (W1015, Promega) and X-ray film (F-BX810, Phenix).

## Co-immunoprecipitation

The co-immunoprecipitation was conducted according to a standard protocol as described previously in ref. [63] with minor modifications. In brief, ALT#1 cells were transduced with vector control or Flag-tagged SENP6$^{C1030A}$. Cell lysates were prepared by lysing cells in lysis buffer (50 mM Tris pH 7.5, 120 mM NaCl, 1% NP-40, 5 mM EDTA supplemented with a protease inhibitor cocktail). After quantification, cell lysates with 500 µg of proteins were incubated with IgG or anti-Flag M2 antibodies overnight at 4 °C. The next day, lysates were incubated with Protein G magnetic beads for 2 h and washed three times using lysis buffer. The captured proteins were eluted by boiling them in a 2x loading buffer containing 200 mM dithiothreitol (DTT). The levels of Flag-SENP6$^{C1030A}$, KDM2A, and PML in input and pull-down samples were then analyzed by western blot.

## Cell cycle synchronization

Cells were synchronized in the G2 phase by sequential treatment of thymidine and CDK1 inhibitor Ro-3306 (S7747, Selleckchem). Briefly,

cells were first cultured in a medium containing 2 mM thymidine (Sigma-Aldrich) for 21 h. After washing twice with PBS followed by once with growth media, the cells were released into fresh medium for 3 h before treatment with 10 mM CDK1 inhibitor for 12 h.

## Flow cytometry analysis (FACS)

Cells collected by trypsin and resuspended in PBS containing 1 mM EDTA were fixed in ice-cold ethanol overnight. Ethanol-fixed single-cell suspensions were stained for DNA analysis with 1% BSA, 0.1% Tween-20, 0.1 mM EDTA, 0.5 mg/mL RNaseA, and 10 μg/mL propidium iodide (PI) in 0.25 ml PBS. Cells were incubated for 30 min at 37 °C and equilibrated at room temperature in the dark for at least 10 min. Cells were analyzed by an LSRII flow cytometer (BD Biosciences). Cell cycle population analysis was conducted with FlowJo v5 software (FlowJo).

## Terminal restriction fragment (TRF) analysis

TRF analysis was conducted as previously described in ref. [22]. In brief, genomic DNA was prepared using Wizard genomic DNA purification kit (Promega) as manufacturer's instruction. For telomere length and Southern blot analysis, genomic DNA (~ 5 μg) was digested with AluI + MboI restriction endonucleases, fractionated in a 0.7% agarose gel, denatured, and transferred onto a GeneScreen Plus hybridization membrane (PerkinElmer). The membrane was cross-linked, hybridized at 42 °C with 5′-end-labeled $^{32}$P-(TTAGGG)$_4$ probe in Church buffer, and washed twice for 5 min each with 0.2 M wash buffer (0.2 M Na$_2$HPO4 pH 7.2, 1 mM EDTA, and 2% SDS) at room temperature and once for 10 min with 0.1 M wash buffer at 42 °C. The images were analyzed by Phosphor-imager, visualized by Typhoon 9410 Imager (GE Healthcare), and processed with ImageQuant 5.2 software (Molecular Dynamics).

## C-circle assay

C-circle assay was performed as described in refs. [21,64] with minor modifications. Briefly, genomic DNA digested with AluI and MboI was cleaned up by phenol-chloroform extraction and precipitation. An aliquot of purified DNA was diluted in nuclease-free water, and concentrations were measured to the indicated quantity (~15 ng/μl) using a Nanodrop spectrophotometer (Thermal Scientific). Sample DNA (30 ng in a total volume of 10 μl) was combined with 10 μl reaction mix [0.2 mg/ml BSA, 0.1% Tween, 0.2 mM each dATP, dGTP, dTTP, 2 × φ29 Buffer, and 7.5 U φDNA polymerase (NEB)]. The reactions were mixed well, incubated at 30 °C for 8 h, and then at 65 °C for 20 min. The reaction products were diluted to 400 μl with 2 × SSC, dot-blotted onto a 2 × SSC-soaked GeneScreen Plus membrane, and hybridized with a $^{32}$P-labeled (CCCTAA)$_4$ probe at 37 °C overnight to detect C-circle amplification products. The blots were washed four times at 37 °C in 0.5 × SSC/0.1% SDS buffer, exposed to Phosphor-imager screens, visualized by Typhoon 9410 Imager (GE Healthcare Life Sciences), and quantified with ImageQuant 5.2 software (Molecular Dynamics).

## Chromatin immunoprecipitation (ChIP) assays

ChIP assays were performed as described previously in ref. [22]. Briefly, cells (~1 × 10$^7$) were cross-linked in 1% formaldehyde with shaking for 15 min, quenched by the addition of glycine to a final concentration of 0.125 M, and lysed in 1 ml SDS lysis buffer (1% SDS, 10 mM EDTA, and 50 mM Tris-HCl, pH 8.0) supplemented with 1 mM PMSF and protease inhibitor cocktails (Sigma-Aldrich). The lysates were sonicated with a Diagenode Bioruptor, cleared by centrifugation to remove insoluble materials, and diluted tenfold into IP Buffer (0.01% SDS, 1.1% Triton X-100, 1.2 mM EDTA, 16.7 mM Tris pH 8.1, 167 mM NaCl, 1 mM PNSF, and protease inhibitors cocktail) for IP reaction at 4 °C overnight. Each immuno-complex was washed five times (1 ml wash, 10 min each) in ChIP-related wash buffer at 4 °C, eluted by the addition of 150 μl Elution buffer (10 mM Tris, pH 8.0, 5 mM EDTA, and 1% SDS) at 65 °C for 30 min, and the elutes were placed at 65 °C for overnight to reverse

cross-linking. The elutes was further treated with Proteinase K in a final concentration of 100 μg/ml at 50 °C for 2 h and ChIP DNA was purified by Quick PCR Purification Kit (Life Technologies) following the manufacturer's instruction. ChIP DNA was denatured, dot-blotted onto GeneScreen Plus blotting membranes (PerkinElmer) and cross-linked at 125 mJ. The Oligonucleotide probe for telomere or Alu repeats was labeled with [$^{32}$P]-ATP (3,000 Ci/mmol) and T4 nucleotide kinase (New England Biolabs). The membrane was hybridized in Church hybridization buffer containing a $^{32}$P-labeled probe at 42 °C overnight, washed three times in 0.04 N Na-phosphate, 1% SDS, 1 mM EDTA at 42 °C, developed with a Typhoon 9410 Imager (GE Healthcare Life Sciences) and quantified with ImageQuant 5.2 software (Molecular Dynamics). Antibodies used in the ChIP assay were anti-Flag (F1804, Sigma), anti-H3K36me2 (2901, Cell Signaling), and mouse IgG (sc2025, Santa Cruz Biotechnology).

## Detection of ALT-directed telomere DNA synthesis

To visualize DNA synthesis at telomeres, synchronized G2 cells were incubated with 20 mM EdU for 2 h. Cells were permeabilized, then fixed with a 4% formaldehyde PBS solution. The Click-iT® Alexa Fluor 555 azide reaction was then performed according to the manufacturer's instructions (Click Chemistry Tools). After washing with 1x PBS, cells were incubated for 60 min in blocking solution (1% BSA, 10% FBS, 0.2% fish gelatin, 0.1% Triton X-100, and 1 mM EDTA in 1x PBS) before PML immuno-staining at RT for 2 h. After incubation of secondary antibodies and three washes with PBST, cells were fixed again with PFA for 10 min at RT, washed in 1x PBS, dehydrated in ethanol series (70, 95, 100%), and air-dried. Coverslips were denatured for 10 min at 85 °C in a hybridization mix [70% formamide, 10 mM Tris-HCl, pH 7.2, and 0.5% blocking solution (Roche)] containing 100 nM telomeric PNA probe TelC-FITC (F1009, PNA Bio), and hybridization was continued for 2 h at room temperature in the dark moisturized chambers. Coverslips were washed three times with Wash solution (70% formamide, 2x SSC) for 10 min each and in 2x SSC, 0.1% tween-20 three times for 5 min each. During the second wash, cells were stained with DAPI. Slides were mounted with VectorShield (Vector Laboratories). Images were captured with a 60x lens on an Olympus FLUO-VIEW laser scanning confocal microscope (Olympus).

## Live cell imaging

H2B-GFP-expressing control or KDM2A targeting sgRNA-transduced cells were seeded at 8 × 10$^4$ cells per well in fibronectin-coated 12-well 1.5 mm glass-bottom wells (Cellvis). Following culture in DMEM medium supplemented with 15% FBS for 24 h, cells were synchronized in the G2 phase by sequential treatment of thymidine and CDK1 inhibitor Ro-3306 (S7747, Selleckchem). Cells were firstly cultured in a medium containing 2 mM thymidine (Sigma-Aldrich) for 24 h. After washing twice with PBS followed by once with growth media, the cells were then released into fresh medium for 2 h before treatment with 10 mM CDK1 inhibitor for 16 h. Finally, cells were washed twice with PBS and once with growth media before being subjected to live cell imaging with Zeiss Cell Observer (ZEISS). Cells were monitored for 8 h at 5 min intervals. Movies are output by Zeiss ZEN software (ZEISS).

## Statistics and reproducibility

Details regarding quantitation and statistical analysis are provided in the figures and figure legends. We determined experimental sample sizes on the basis of preliminary data. All results are expressed as mean ± s.e.m. GraphPad Prism software (version 7.0e) was used for all statistical analysis. The normal distribution of the sample sets was determined before applying Student's two-tailed $t$-test for two group comparisons. Differences were considered significant when $P < 0.05$. For western blotting micrographs, the experiments were repeated three times independently with similar results, and representative images/blots are shown.

## Reporting summary

Further information on research design is available in the Nature Portfolio Reporting Summary linked to this article.

## Data availability

The data that support the findings of this study are available within the article, Supplementary Information, or Source Data file. All the uncropped western blots and raw data are provided as a Source data file. Source data are provided with this paper.

## Code availability

There was no new code generated for this study.

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

## Acknowledgements

This work was partly supported by a research award from the William Rhodes and Louise Tilzer-Rhodes Center for Glioblastoma at NewYork-Presbyterian Hospital. Additional funding was provided to J. Paik by NIH/NCI grant P01 #CA214274. J-Y. Jang is partly supported by Basic Science Research Program through the National Research Foundation of Korea (NRF), funded by the Ministry of Education (2021R1A6A3A03039136). H. Zheng was partly supported by the Sontag Foundation and the Chen & Xiao AntiCancer Foundation.

## Author contributions

F.L., Y.W., I.H., J.-Y.J., L.X., Z.D., E.Y.Y., Y.C., C.W., Z.H., Y.-H.H., X.H., and L.Z. carried out experiments. J.Y. performed the sequencing data analysis. N.F.L., P.M.L., H.Y., J.P., and H.Z. supervised this study. F.L., J.P., and H.Z. analyzed the data and wrote the manuscript.

## Competing interests

The authors declare no competing interests.
