## [Peer Review File · Nature Communications]

Reviewers' Comments:

Reviewer #1:

Remarks to the Author:

In this study, Li et al., demonstrate an essential role for the histone H3K36 demethylase KDM2A in the cytoprotection of ALT cancer cells. The authors used an isogenic pair of fibroblasts in which ALT was induced following the deletion of the chromatin remodeling factor ATRX – which is often mutated in cancer cells that activate ALT, as well as corresponding control lines expressing telomerase. They then conducted a CRISPR screen to identify essential genes that are necessary for the survival of the ATRX-deficient ALT cancer cells.

Their library of sgRNAs comprised ~5,000 gRNAs that target 455 chromatin modifier genes. Of those, the authors identify the histone demethylase KDM2A as a new factor that is crucial for the survival of ATRX-deficient ALT cancer cells. The dependency of ALT cancer cells on this factor is rigorously assessed in panels of cancer cell lines. The authors proceed to show that this dependency relies on, not unexpectedly, the enzymatic (i.e., histone demethylase) and histone modification binding properties associated with the JmjC and PHD finger domains of KDM2A. Yet, curiously the authors provide evidence that ATRX loss is not the only cause of the enhanced sensitivity of cells that activate ALT.

In functional studies, the authors demonstrate that KDM2A deficiency causes increased H3K36me2 at telomeres in ATRX-KO ALT cells. Curiously, KDM2A deficiency does not appear to have a discernable impact on telomere maintenance by ALT. Specifically, by examining several markers of ALT, the authors failed to observe alterations in the frequency of ALT-associated APBs, telomere DNA synthesis (C-circles & EdU experiments), and ultimately no effect on telomere length. The authors did observe evidence of mitotic defects, specifically in the form of abnormal segregation of chromosomes. From these observations, the authors hypothesize that KDM2A acts after DNA synthesis and serves to mediate the disentanglement of telomeric clusters prior to chromosome segregation in mitosis that may be mediated by BLM helicase. Furthermore, these mitotic defects are then linked with defects in the recruitment of the SUMO-protease SENP6 which was also recovered as having a link with KDM2A from a focused CRISPR KO screen. The authors show that SENP6 recruitment to telomere is impaired in KDM2A KO cells.

Overall, the study has several interesting features. Uncovering KDM2A as a possible protein of interest in ALT is quite interesting. The sensitivity analyses are thorough, albeit still lacking. Similarly, the functional work on ALT seems to be inconclusive as does the final section in which the SENP6 angle is explored. My overall sense at the end of this paper is that there are a series of observations made, but we are not much wiser as to why KDM2A has this elevated role in ATRX KO ALT cells. There are many unanswered questions, any one of which should be further examined in greater detail to bring some clarity to the study. An additional overall comment is that there are lots of figures. However, the placement and connection among these is at times disparate and confusing. I would suggest that a more linear approach be taken. In addition, I find that the titles of the subsections do not support the outcomes of the experiments and could be construed as misleading.

Here are my focused critiques that I would suggest the author's address (in order of figures):

Figure 1-2.

- This is a comprehensive analysis. Structurally, this figure would benefit from a western validating the ATRX status of the ATRX-KO cells rather than, or in addition to, the 53BP1 IF. The latter seems out of place and more connected to 4C.
- Suggest adding Figure S1B to main Figure 1. Were other known ATRX-synthetic lethals (HIRA-ASF1 complex) identified in 2 studies and other ALT synthetic lethals included as controls (BLM, FANCM?) uncovered?
- What day are the images of the crystal violets taken? It is not indicated in the legend.
- Rescue experiments with wt-ATRX should be added to these panels. This relates to Fig 4 also.
- Can normal untransformed parental cell lines be included in this analysis?
- Please add statistics to each of the graphs.

Figure 3.

- The conclusions (stated in the heading of this section) and the data are discordant. The data appears to show that the JmjC, CXXC, PHD and LRR repeats are each required. Unless I have misunderstood this.
- Experiments with a LRR repeat mutant should be included if the latter is true.
- Figure 3D – do these mutants affect H3K36me1/2?

Figure 4.

- This section is entitled KDM2A functions in ALT-directed telomere maintenance. The legend is then entitled ATRX alone is insufficient to confer KDM2A dependency. I would contend that this is not what the figure shows - at least not conclusively. At best these titles are confusing. Rather, here more evidence of growth defects and selectivity for ALT is shown. Furthermore, there are efforts made here to show that KDM2A localizes to telomeres in ALT cells. A proteomics study is cited, and ChIP results are shown. The ChIP shown in 4G is lacking controls i.e., Alu repeats should be included. Data for H3K36me2 shown in 4J should be normalized to histone H3, no? It would be ideal to show H3K36me2 ChIP results from experiments in multiple cell lines. Furthermore, is ATRX-KO sufficient to recruit KDM2A to telomeres?

Figures 5-6

- The data that would relate to ALT telomere maintenance are shown in Figure 5. Yet there really are no data to support that KDM2A has a role in ALT telomere maintenance. It would be helpful and more thorough to present data from multiple ATRX-KO and ATRX WT ALT cell lines. The rationale for scoring cells with 4 APBs is not explained. The telomere restriction analysis does not have adequate resolution to correctly assess telomere length in ALT cancer cells. The same applies to Supplementary Figure 5e.
- Considering the defects in telomere declustering – have the authors detected evidence of increased DNA synthesis in mitosis (MiDaS)?
- Can the authors show that the mitotic defects can be rescued with WT-KDM2A and/or ATRX.
- Can the authors assess the mechanism of cell death in KDM2A-KO ATRX-deficient cells?

Figure 7

- While the connection between KDM2A and SENP6 is extremely interesting, the analysis here is of global SUMOylation rather than direct analysis of a relationship between the 2 proteins and de-SUMOylation. Is KDM2A targeted for SENP6 de-SUMOylation? Is the enzymatic or chromatin binding of KDM2A necessary for SENP6 recruitment?
- The model presented is very vague and lacks a conceptual advance.

Reviewer #2:

Remarks to the Author:

Histone demethylase KDM2A is a selective vulnerability of cancers relying on alternative telomere maintenance

Fei Li et al

ALT activity is required in human cancers that lack TERT activity, including some osteosarcoma, leiomyosarcoma, liposarcoma, glioblastoma, and neuroendocrine pancreatic cancers. The specific activation of ALT in tumors makes it a potential target for therapy. With the recent insights into the ALT pathway, several strategies have been proposed to exploit the dependency of tumors on ALT. However, there are currently no successful target therapy. In this paper, they developed isogenic pairs of ALT cell lines from human lung fibroblast. Using the pairs of these cells they performed CRISPR-based pooled genetic screens with protocols minimizing the noises. In this paper, they screened 455 epigenetic modifiers, and identified several genes, ASM21, KDM2A, KMT5B, RNF8 and SETDB1, which were specifically required for survival of ALT-dependent cells. They focused on KDM2A for further analysis because KDM2A showed the strongest relation to ALT-cell survival. Further, they showed that KDM2A was required for disassembly of ALT-specific recombinogenic multitelomere bodies after homology-directed telomere DNA synthesis. Finally, they suggested that this de-clustering was mediated through KDM2A-controlling the multitelomere localization of SENP6 that executed SUMO deconjugation of telomere-associated proteins. Their

results provide important clues for therapy targeting ALT-.

They suggested that KDM2A worked on telomere, and was directly involved in the process. However, they did not detect KDM2A protein in most of the experiments. According to Fig. 7a, the localization of SMC5 in siCtrl cells was changed by KDM2A KO, though the level of SMC5 protein was not affected (Fig 7c). These results may suggest that KDM2A KO affected the machine structure for ALT. Therefore, it is possible that there are other ways, in which KDM2A works indirectly on the ALT controls. For example, KDM2A controls the expression of genes involved in ALT de-clustering. Requirement of CxxC domain and JmjC domain for proliferation of ALT-cells is consistent to the hypothesis that KDM2A contributes to ALT through controlling the gene expressions. Then I recommend they would discuss also at least the second possibility, in which KDM2A controls the gene expressions involved in ALT.

Specific comments:

p. 13, line 280: As a structure-specific histone H3K36-demethylase, KDM2A consists of a zinc finger CXXC domain that constrains its recruitment to nucleosomes surrounded by unmethylated linker DNA regions. Interestingly, the mammalian telomere DNA repeats are hypomethylated due to their lack of CpG-dinucleotide sequences.

Comment: The CxxC domain binds to unmethylated CpG sequences. As they stated that the mammalian telomere DNA repeats lack of CpG-dinucleotide sequences, it was questionable that KDM2A specifically bound to telomere sequences. Any experiments are required to show if KDM2A preferentially bind to telomere sequences by, for example, the gel shift experiments with comparing appropriate control DNA sequences.

p. 13, line 288: Moreover, telomere dot-blot analysis of KDM2A-depleted Saos2 cells indicated a significantly increased level of telomere H3K36me2 as compared to the sgCtrl-transduced cells (Fig. 4i-k).

Comment: FlagKDM2A strongly bound to telomere in ALT#1 cells compared to Saos2 cells. It is recommended to compare the levels of H3 and H3K36me2 in ALT#1 cells with FlagKDM2A expression to the cells without FlagKDM2A expression.

p. 15, line 343: The immuno-FISH analysis of the G2 phase-synchronized cells revealed comparable levels of APB formation in the control and KDM2A depleted ALT#1 cells (Fig. 6a, b), indicating that KDM2A loss did not affect telomere clustering and APB assembly.

Comment: The results of Fig. 6 were clear. To clarify the roles of KDM2A, detection of KDM2A protein in this experimental context is recommended.

p. 17, line 386: Moreover, SMC5 siRNA treatment in sgK#1-transduced ALT#1 cells significantly prevented the abnormal M-phased multitelomere cluster formation as compared to those treated with scrambled control siRNA (Fig. 7a-c).

Comment: According to Fig. 7a, the localization of SMC5 was changed by KDM2A KO, though the amount of SMC5 protein was not affected (Fig 7c). Did KDM2A affect SMC5 localization?

p. 18, line 388: Moreover, SMC5 siRNA treatment in sgK#1-transduced ALT#1 cells significantly prevented the abnormal M-phased multitelomere cluster formation as compared to those treated with scrambled control siRNA (Fig. 7a-c). These results indicate that KDM2 indeed acts downstream of SMC5/6.

Comment: Because RNAi knock-down of SMC5 blocked the recruitment of telomeres to PML bodies and significantly reduced multitelomere cluster foci formation (Supplementary Fig. 7a, b), reduction of the abnormal M-phased multitelomere cluster formation would simply reflect the reduced levels of multitelomere cluster foci formation by SMC5 KD. Indeed, the difference between the values of siCtrl and siSMC5 in sgCtrl cells seemed to be similar and the difference between the values of siCtrl and siSMC5 in sgK#1 cells.

p. 19, line 432: Analysis of the G2-synchronized ALT#1 cells found that the Flag-tagged SENP6C1030A mutant was indeed localized to clustered telomeres (Fig. 7k).

Comment: In sgCtrl cells, the dotted Flag signals were detected, most of which were overlapped with telomere foci. In the sgK#1 cells, there was one circle that was not overlapped with telomere foci. Was this circle overlapped with a signal for PML? In the Figure 5b PML were detected as dots overlapping telomere signals in sgCtrl and sgK#1 cells. If not, did the expression of SENP6C1030A

mutant disrupt PML in the context of KDM2A KO ALT1 cells? Please detect PML in Figure 7k. It is also recommended to detect KDM2A protein in Fig. 7k.

When they claim the direct involvement of KDM2A in ALT, it is required to show if SENP-6 directly binds to KDM2A or SENP-6 directly binds to H3K36, depending on the methylation status.

p. 22, line 501; p.23, line 512-521: Our results further indicate that the process is regulated by KDM2A-directed telomere histone H3K36me2 demethylation.

Comment: There was few evidences showing that KDM2A-directed telomere histone H3K36me2 demethylation functions in ALT. The results in Figure 3 showed that KDM2A demethylase activity and DNA binding activity were involved in the ALT process. These activities may be involved in controlling the expressions of the genes required for ALT de-clustering process.

Reviewer #3:

Remarks to the Author:

I read the study by Li et al, "Histone demethylase KDM2A is a selective vulnerability of cancers relying on alternative telomere maintenance" with great interest. In brief, the authors used a targeted screening approach to identify that KDMA2 deletions results in specific lethality of ALT cells. Through directed molecular biology, they identify loss of KDMA2 in ALT cells impairs the ability of cells to detangle the phase-separated telomere clusters (i.e. APBs) that are known to occur in cells of this type. They identified KDMA2 functions upstream of SENP6, which is required to deSUMOylate proteins within APBs. Loss of KDMA prevents SENP6 recruitment, and the continued entanglement of presumed recombination intermediates in mitosis leads to mitotic catastrophe.

Overall, the study is of excellent quality and represents a novel approach to identifying putative vulnerabilities in ALT cancers. I particularly appreciated the use of varied and well-designed CRISPR screens. Supporting experiments were largely well done and informative to the study. I find the report strong and believe it is an excellent candidate for publication once the authors address the concerns I have below. I stress these are largely minor concerns and expect they can be handled by the research team. Congratulations to the authors on an excellent study.

1. Line 125: How did the authors identify 1,000 cells per sgRNA coverage?
2. Line 140: Describing the spiked in controls would help orient the reader.
3. Line 336: Is there a reference for cell-cycle-dependent PML-NB or APB dissociation?
4. Line 353: This line is somewhat misleading. TRF1 does not dissociate from mitotic telomeres and can readily be seen on mitotic telomeres (e.g. with GFP-TRF1). However these telomere foci are dim compared to the very bright APBs being visualized. This should be re-worded to better convey.
5. Mitotic catastrophe is a very poorly defined phenomena. What specifically is occurring? Solely chromosome segregation errors, or is it also mitotic death (a known consequence of lethal replication stress, PMID: 31530811).
6. Discussion: The description of "synthetic lethality" within the ALT context is difficult as KDM2A is not synthetic lethal with a second gene but is instead 'synthetic lethal' with a phenotype (i.e. ALT). The Authors show indeed KDM2A is not synthetic lethal with ATRX. IT might be worth clarifying for the reader.
7. Figure 2 – these data would benefit by tarking KDM2A in matched cancer types of the ALT-cancer cells used in this study. I.e. a telomerase positive osteosarcoma line (to match with U2OS and Saos2)
8. The movies, while lovely to watch, lack significant and important details. There are no time stamps to indicate time, there are no scale bars to indicate size. Further it is not clear which events are important. The authors need to include time stamps and scale bars, and it may be better to identify individual events with arrows or other markers to draw the readers attention.

Minor comments

9. The imaging related to telomere condensates being maintained in mitosis in the KDM2A is striking. While not critical to this study, should the authors be able to visualize this with live

imaging it would be a powerful tool.

Detailed response to recommendations

We are grateful to the reviewers for their thorough and thoughtful critiques and appreciate the opportunity to submit a revised manuscript.

Reviewer #1 - ALT in cancer (Remarks to the Author):

In this study, Li et al., demonstrate an essential role for the histone H3K36 demethylase KDM2A in the cytoprotection of ALT cancer cells. The authors used an isogenic pair of fibroblasts in which ALT was induced following the deletion of the chromatin remodeling factor ATRX – which is often mutated in cancer cells that activate ALT, as well as corresponding control lines expressing telomerase. They then conducted a CRISPR screen to identify essential genes that are necessary for the survival of the ATRX-deficient ALT cancer cells.

Their library of sgRNAs comprised ~5,000 gRNAs that target 455 chromatin modifier genes. Of those, the authors identify the histone demethylase KDM2A as a new factor that is crucial for the survival of ATRX-deficient ALT cancer cells. The dependency of ALT cancer cells on this factor is rigorously assessed in panels of cancer cell lines. The authors proceed to show that this dependency relies on, not unexpectedly, the enzymatic (i.e., histone demethylase) and histone modification binding properties associated with the JmjC and PHD finger domains of KDM2A. Yet, curiously the authors provide evidence that ATRX loss is not the only cause of the enhanced sensitivity of cells that activate ALT.

In functional studies, the authors demonstrate that KDM2A deficiency causes increased H3K36me2 at telomeres in ATRX-KO ALT cells. Curiously, KDM2A deficiency does not appear to have a discernable impact on telomere maintenance by ALT. Specifically, by examining several markers of ALT, the authors failed to observe alterations in the frequency of ALT-associated APBs, telomere DNA synthesis (C-circles & EdU experiments), and ultimately no effect on telomere length. The authors did observe evidence of mitotic defects, specifically in the form of abnormal segregation of chromosomes. From these observations, the authors hypothesize that KDM2A acts after DNA synthesis and serves to mediate the disentanglement of telomeric clusters prior to chromosome segregation in mitosis that may be mediated by BLM helicase. Furthermore, these mitotic defects are then linked with defects in the recruitment of the SUMO-protease SENP6 which was also recovered as having a link with KDM2A from a focused CRISPR KO screen. The authors show that SENP6 recruitment to telomere is impaired in KDM2A KO cells.

Overall, the study has several interesting features. Uncovering KDM2A as a possible protein of interest in ALT is quite interesting. The sensitivity analyses are thorough, albeit still lacking. Similarly, the functional work on ALT seems to be inconclusive as does the final section in which the SENP6 angle is explored. My overall sense at the end of this paper is that there are a series of observations made, but we are not much wiser as to why KDM2A has this elevated role in ATRX KO ALT cells. There are many unanswered questions, any one of which should be further examined in greater detail to bring some clarity to the study. An additional overall comment is that there are lots of figures. However, the placement and connection among these is at times disparate and confusing. I would suggest that a more linear approach be taken. In addition, I find that the titles of the subsections do not support the outcomes of the experiments and could be construed as misleading.

We thank the referee for the thoughtful review and critiques. We have been able to address the comments in our revised manuscript both by presenting significant new data in the main and supplementary figures and by providing clarification.

Here are my focused critiques that I would suggest the author's address (in order of figures):

Figure 1-2.

- This is a comprehensive analysis. Structurally, this figure would benefit from a western validating the ATRX status of the ATRX-KO cells rather than, or in addition to, the 53BP1 IF. The latter seems out of place and more connected to 4C.

Thanks for the suggestion. A western blot analysis of ATRX protein expression in control and ATRX-KO ALT cells is now included in the manuscript as Figure 1a. The 53BP1/telomere immuno-FISH studies of control and ALT cells have been moved to Supplementary section as Figures S1a and 1b.

- Suggest adding Figure S1B to main Figure 1. Were other known ATRX-synthetic lethals (HIRA-ASF1 complex) identified in 2 studies and other ALT synthetic lethals included as controls (BLM, FANCM?) uncovered?

Following the suggestion, the screen heatmap depicting log₂FC of sgRNA abundance of selected genes (previous Figure S1B) is now incorporated into the main figure panel as Figure 1d.

Although HIRA, ASF1, BLM, and FANCM are not in the gene list of chromatin modifier-targeting sgRNA library of the current study, they were included in our other yet published library screen studies. According to those screen results, only FANCM is selectively essential to ALT cells (Reviewer Figure 1). By comparison, BLM and HIRA are required for the growth of both control IMR90-T and ALT cells. With respect to ASF1, we found that depletion of ASF1A had a minimal growth effect on either control IMR90-T or ALT cells, probably due to the redundant function shared by its paralog ASF1B gene.

Reviewer Figure 1. Heatmap depicting gene dependency score (GDS) of the indicated genes in IMR90-T (T#1 and T#2), ALT#1, ALT#2, or ALT#3 cells. The GDS score was calculated by averaging the log₂FC of all sgRNAs targeting that gene after 16 population doublings.

- What day are the images of the crystal violets taken? It is not indicated in the legend.

Thanks for pointing it out. The information is now added to the legends of the revised manuscript.

- Rescue experiments with wt-ATRX should be added to these panels. This relates to Fig 4 also.

As requested by the reviewer, we have investigated the impact of ectopic ATRX expression in ATRX-null U2OS cells. Our data show that ATRX re-expression in U2OS cells not only abrogated ALT-associated PML body formation, it also significantly attenuated their KDM2A dependency as indicated by the completion-based proliferation assay (incorporated into the manuscript as Supplementary Figures S3a-d). In addition to the new data on ATRX re-expression, we also present DAXX rescue experiments in G292, an ALT cell line that harbors genetic alteration at *DAXX*. Consistent with our finding that KDM2A is selectively required for ALT cells, we found that transduction of wild-type DAXX into G292 cells abolished APB formation and mitigated their KDM2A dependency. These new data are incorporated into the revised manuscript as Supplementary Figure S3e-h.

- Can normal untransformed parental cell lines be included in this analysis?

As per the reviewer's request, we have conducted a competition-based proliferation assay in the untransformed IMR90 cells. The new data is now included as Supplementary Figure S4f.

- Please add statistics to each of the graphs.

Thanks for the suggestion. We have added missing statistical values to the graphs.

Figure 3.

- The conclusions (stated in the heading if this section) and the data are discordant. The data appears to show that the JmjC, CXXC, PHD and LRR repeats are each required. Unless I have misunderstood this.

We have modified the heading of this section as suggested.

- Experiments with a LRR repeat mutant should be included if the latter is true.

Following the reviewer's suggestion, we have constructed a CRISPR-resistant KDM2A LRR deletion mutant (Δ LRR, aa 1-aa 945). A competition-based proliferation assay was then conducted in Saos2 cells complemented with KDM2A wild-type or Δ LRR mutant. Consistent with the exon-tilling results, the assay shows that LRR domain of KDM2A is required for its ALT supporting function. The data are included in the revised manuscript as Supplementary Figure S6a and b.

- Figure 3D – do these mutants affect H3K36me1/2?

As per the reviewer's request, we have examined the levels of H3K36me1/2 and H3 using histone acid extraction prepared from Saos2 cells that were transduced with vector control, KDM2A wild-type, or indicated mutants. The western blot analysis found that cells transduced with wild-type KDM2A or HP1 protein interaction mutant (V801A/L803A), but not mutants defective of DNA binding (S603D), PHD domain structural integrity (C620/623A) or demethylase activity (D214A or N298A), showed attenuated cellular H3K36me2 levels as compared to the cells transduced with vector control. With respect to H3K36me1 levels, we did not notice a significant difference among these cells. The new data is incorporated into Figure 3d.

Figure 4.

- This section is entitled **KDM2A functions in ALT-directed telomere maintenance. The legend is then entitled ATRX alone is insufficient to confer KDM2A dependency. I would contend that this is not what the figure shows - at least not conclusively. At best these titles are confusing. Rather, here more evidence of growth defects and selectivity for ALT is shown. Furthermore, there are efforts made here to show that KDM2A localizes to telomeres in ALT cells. A proteomics study is cited, and ChIP results are shown. The ChIP shown in 4G is lacking controls i.e., Alu repeats should be included. Data for H3K36me2 shown in 4J should be normalized to histone H3, no? It would be ideal to show H3K36me2 ChIP results from experiments in multiple cell lines. Furthermore, is ATRX-KO sufficient to recruit KDM2A to telomeres?**

Following the reviewer's suggestion, we have modified the heading and legend.

The ChIP/dot blot of Alu repeats and quantification (serving as the controls for Figure 4g) are added into the revised manuscript as Supplementary Figures S8a and b.

The quantification for H3K36me2 shown in Figure 4j has now been normalized to histone H3.

Following the reviewer's suggestion, we also conducted anti-H3K36me2, anti-H3, and IgG ChIP/ telomere dot blot in sgCtrl or sgK#1-transduced ALT#1 cells. The new data are now included as Supplementary Figures S8c-e.

Furthermore, we have performed anti-Flag or IgG ChIP/telomere dot blot analysis in control IMR90-T and ATRX-deleted dATRX#1 cells that were transduced with Flag-KDM2A. The data show that ATRX-KO in IMR90-T cells marginally enhanced KDM2A interaction with telomeres [dATRX#1: 0.247% \pm 0.023% vs. IMR90-T: 0.202% \pm 0.037% of telomeric inputs; P = 0.033] (Reviewer Figure 2a-c).

Reviewer Figure 2. ATRX deletion marginally enhances KDM2A interaction to telomeres in IMR90-T cells. a Western blot analysis of Flag and ACTB in control or Flag-KDM2A-transduced IMR90-T or dATRX#1 cells. **b, c** Telomere dot blot analysis (**b**) and quantification (**c**) of anti-Flag or IgG ChIP. The relative enrichment was calculated after normalization of ChIP DNA signals to the respective input DNA signals. Means \pm s.e.m.; N = 3; paired t-test.

Figures 5-6

- The data that would relate to ALT telomere maintenance are shown in Figure 5. Yet there really are no data to support that KDM2A has a role in ALT telomere maintenance. It would be helpful and more thorough to present data from multiple ATRX-KO and ATRX WT ALT cell lines. The rationale for scoring cells with 4 APBs is not explained. The telomere restriction analysis does not have adequate resolution to correctly assess telomere length in ALT cancer cells. The same applies to Supplementary Figure 5e.

Following the reviewer's suggestion, we have conducted new experiments to investigate the impact of KDM2A depletion on two other ALT cell lines - ALT#2 (ATRX-KO ALT cells) and G292 cells (ATRX-WT ALT cells). Consistent with the original findings that were derived from ALT#1 and Saos2 cells, depletion of KDM2A in ALT#2 or G292 cells impaired mitotic ALT telomere de-clustering without significantly affecting their G2-phase telomere clustering and APB formation. These new data are now included as Supplementary Figure S11e-g (for ALT#2 cells) and Supplementary Figure S11h-j (for G292 cells).

As regarding to the APB formation in sgCtrl and sgK#1-transduced ALT#1 cells, we have re-analyzed the immune-FISH images by counting the numbers of APBs in individual cells. The new quantification data for ALT#1 and Saos2 cells are incorporated into Figure 5c and Supplementary Figure S9d respectively.

Following the reviewer's suggestion, we have also repeated the telomere restriction analysis in ALT#1 and Saos2 cells that were transduced with sgCtrl, sgK#1 or sgK#2. The new data are now included as Figure 5d (for ALT#1 cells) and Supplementary Figure S9f (for Saos2 cells).

- Considering the defects in telomere declustering – have the authors detected evidence of increased DNA synthesis in mitosis (MiDaS)?

To address this question, we have conducted new experiments to analyze mitotic DNA synthesis (MiDaS) in sgCtrl and sgKDM2A-transduced ALT#1 cells. The data show that KDM2A depletion did not significantly affect the telomeric MiDaS of ALT cells (Reviewer Figure 3a and b).

Reviewer Figure 3. KDM2A depletion does not significantly affect telomeric MiDaS in ALT cells. **a** Representative images of EdU incorporation at metaphase telomeres of sgCtrl or sgK#1-transduced ALT#1 cells. Cells were synchronized in the G2 phase with the CDK1 inhibitor RO3306 before released into mitosis for 60 minutes in the presence of EdU and colcemid (0.1 µg/mL). Mitotic cells were collected for metaphase preparation. **b** Quantification of metaphases with EdU incorporation at telomeres. Means ± s.e.m.; N = 3; unpaired t-test.

- Can the authors show that the mitotic defects can be rescued with WT-KDM2A and/or ATRX.

To address this question, we have conducted reconstitution experiments. As expected, the new data show that ectopic transduction of ALT#1 cells with a sgK#1-resistance Flag-KDM2A(r) or an ATRX cDNA construct was capable of rescuing the KDM2A deficiency-induced mitotic defects. These data are now incorporated into the revised manuscript as Supplementary Figure S11k-m (KDM2A reconstitution) and Supplementary Figure S12a-c (ATRX reconstitution).

- Can the authors assess the mechanism of cell death in KDM2A-KO ATRX-deficient cells?

To address the reviewer's question, we have conducted western blot analysis of the control and KDM2A-KO IMR90-T and ALT#1 cells and observed an increased level of cleaved PARP1 specifically in KDM2A-depleted ALT#1 cells. Further analysis of the mitotic outcomes of the control and KDM2A-KO ALT#1 cells with live-cell imaging revealed a strong mitotic death phenotype in the KDM2A-depleted ALT cells, suggesting that KDM2A depletion induces an apoptosis-dependent mitotic death in ALT cells. These data are now included in the revised manuscript as Supplementary Figure S10e-g.

While our results pinpoint telomere declustering defect as the cause of mitotic catastrophe and cell death in KDM2A deficient ALT cells, it is worthy to note that the detailed molecular mechanism(s) underlying this

process still remains to be defined. Moreover, although mitotic catastrophe has been described for some time, the pathways that regulate mitotic death are less understood (Galluzzi et al., Cell Death Differ. 2018; 25, 486–541; Masamsetti et al., Nat Commun. 2019 Sep 17;10(1):4224). We feel that a rigorous exploration of the detailed molecular mechanism would represent an additional large-scale study and require a very significant amount of time. Therefore, while we are certainly interested in pursuing that direction in the future, we hope that the reviewer will agree that those are beyond the scope of current study.

Figure 7

- While the connection between KDM2A and SENP6 is extremely interesting, the analysis here is of global SUMOylation rather than direct analysis of a relationship between the 2 proteins and de-SUMOylation. Is KDM2A targeted for SENP6 de-SUMOylation? Is the enzymatic or chromatin binding of KDM2A necessary for SENP6 recruitment?

To address the reviewer's questions, we firstly analyzed KDM2A SUMOylation in the control and SENP6 depleted ALT#1 cells. The data revealed no significant change in KDM2A SUMOylation following SENP6 depletion (Reviewer Figure 4), suggesting that KDM2A is unlikely a direct target of SENP6-mediated de-SUMOylation.

Reviewer Figure 4. SENP6 depletion does not affect SUMOylation of KDM2A protein. HA-SUMO2-expressing ALT#1 cells were transduced with sgCtrl, sgSENP6#1 or sgSENP6#2. SUMOylated proteins were captured by anti-HA immunoprecipitation under denaturing condition. Levels of KDM2A, SENP6, ACTB and SUMO2 in inputs and anti-HA pull-downs were analyzed by western blot.

To determine whether the enzymatic and chromatin binding activities of KDM2A are necessary for promoting telomeric SENP6 recruitment in ALT cells, we conducted rescue experiments in Flag-SENP6^{C1030A}-transduced and KDM2A-deficient ALT#1 cells. As expected, immuno-FISH analysis of telomere-SENP6 association revealed significantly higher telomeric SENP6 recruitment in cells complemented with KDM2A wild-type than those with mutants defective of either demethylase (D214A) or chromatin binding (S603D), indicating that the enzymatic and chromatin binding activities of KDM2A are required for promoting telomeric SENP6 recruitment in ALT cells. These data are now included as Supplementary Figure S15a-c.

- The model presented is very vague and lacks a conceptual advance.

Following the suggestion, we have modified the model. The modified one is now incorporated into Fig. 7m.

Reviewer #2 - KDM2A (Remarks to the Author):

Histone demethylase KDM2A is a selective vulnerability of cancers relying on alternative telomere maintenance

Fei Li et al

ALT activity is required in human cancers that lack TERT activity, including some osteosarcoma, leiomyosarcoma, liposarcoma, glioblastoma, and neuroendocrine pancreatic cancers. The specific activation of ALT in tumors makes it a potential target for therapy. With the recent insights into the ALT pathway, several strategies have been proposed to exploit the dependency of tumors on ALT. However, there are currently no successful target therapy. In this paper, they developed isogenic pairs of ALT cell lines from human lung fibroblast. Using the pairs of these cells they performed CRISPR-based pooled genetic screens with protocols minimizing the noises. In this paper, they screened 455 epigenetic modifiers, and identified several genes, ASM21, KDM2A, KMT5B, RNF8 and SETDB1, which were specifically required for survival of ALT-dependent

cells. They focused on KDM2A for further analysis because KDM2A showed the strongest relation to ALT-cell survival. Further, they showed that KDM2A was required for disassembly of ALT-specific recombinogenic multitelomere bodies after homology-directed telomere DNA synthesis. Finally, they suggested that this de-clustering was mediated through KDM2A-controlling the multitelomere localization of SENP6 that executed SUMO deconjugation of telomere-associated proteins. Their results provide important clues for therapy targeting ALT-.

They suggested that KDM2A worked on telomere, and was directly involved in the process. However, they did not detect KDM2A protein in most of the experiments. According to Fig. 7a, the localization of SMC5 in siCtrl cells was changed by KDM2A KO, though the level of SMC5 protein was not affected (Fig 7c). These results may suggest that KDM2A KO affected the machine structure for ALT. Therefore, it is possible that there are other ways, in which KDM2A works indirectly on the ALT controls. For example, KDM2A controls the expression of genes involved in ALT de-clustering. Requirement of CxxC domain and JmjC domain for proliferation of ALT-cells is consistent to the hypothesis that KDM2A contributes to ALT through controlling the gene expressions. Then I recommend they would discuss also at least the second possibility, in which KDM2A controls the gene expressions involved in ALT.

We thank the reviewer for the thoughtful comments. We have added discussion of the other possibilities into the revised manuscript as suggested.

Specific comments:

p. 13, line 280: As a structure-specific histone H3K36-demethylase, KDM2A consists of a zinc finger CXXC domain that constrains its recruitment to nucleosomes surrounded by unmethylated linker DNA regions. Interestingly, the mammalian telomere DNA repeats are hypomethylated due to their lack of CpG-dinucleotide sequences.

Comment: The CxxC domain binds to unmethylated CpG sequences. As they stated that the mammalian telomere DNA repeats lack of CpG-dinucleotide sequences, it was questionable that KDM2A specifically bound to telomere sequences. Any experiments are required to show if KDM2A preferentially bind to telomere sequences by, for example, the gel shift experiments with comparing appropriate control DNA sequences.

Our ChIP/dot blot experiment revealed that KDM2A preferentially binds to the telomeres of ALT#1 and Saos2 cells as compared to non-ALT HeLa cells (Figure 4g and h). Notably, this finding is consistent with the previous proteomic study that also found preferential interaction of KDM2A with ALT telomeres over non-ALT telomeres (Garcia-Exposito et al., Cell Rep, 2016; 17, 1858-1871). Moreover, our anti-H3K36me2 ChIP/telomere dot blot analysis in KDM2A-depleted ALT#1 (Supplementary Figure 8c and d) and Saos2 cells (Figure 4i and j) both showed significant increase of telomere H3K36me2 as compared to their respective control cells. Based on those findings, we conclude that KDM2A may be directly involved in ALT telomere maintenance. However, we are well aware that besides telomeres, KDM2A can also be recruited to numerous other chromosome loci, including CpG islands. Therefore, it was never our intention to suggest that KDM2A only binds to telomeres. We have modified the text to avoid the confusion. Moreover, since KDM2A is a nucleosome interacting protein rather than a DNA binding protein (Bartke et al., Cell. 2010 Oct 29;143(3):470-84; Blackledge et al., Mol Cell. 2010 Apr 23;38(2):179-90; Borgel et al., Nucleic Acids Res. 2017 Feb 17;45(3):1114-1129), we hope the reviewer will agree with us that it is technically not possible to conduct gel shift experiments.

p. 13, line 288: Moreover, telomere dot-blot analysis of KDM2A-depleted Saos2 cells indicated a significantly increased level of telomere H3K36me2 as compared to the sgCtrl-transduced cells (Fig. 4i-k).

Comment: FlagKDM2A strongly bound to telomere in ALT#1 cells compared to Saos2 cells. It is recommended to compare the levels of H3 and H3K36me2 in ALT#1 cells with FlagKDM2A expression to the cells without FlagKDM2A expression.

Following the reviewer's suggestion, we have examined the levels of H3 and H3K36me2 in ALT#1 cells after vector control or Flag-KDM2A transduction. The western blot analysis of the histone acid extracts prepared from Flag-KDM2A-transduced ALT#1 cells revealed a reduced H3K36me2 level as compared to extracts prepared from the vector control-transduced cells (Reviewer Figure 5).

Reviewer Figure 5. Ectopic KDM2A expression reduces H3K36me2 in ALT#1 cells. ALT#1 cells were transduced with vector control or 3xFlag-KDM2A. Histone proteins were prepared from indicated cells by acid extraction. Levels of 3xFlag-KDM2A, total KDM2A, ACTB (loading control), H3K36me2, and total H3 were analyzed by western blot.

p. 15, line 343: The immuno-FISH analysis of the G2 phase-synchronized cells revealed comparable levels of APB formation in the control and KDM2A depleted ALT#1 cells (Fig. 6a, b), indicating that KDM2A loss did not affect telomere clustering and APB assembly.

Comment: The results of Fig. 6 were clear. To clarify the roles of KDM2A, detection of KDM2A protein in this experimental context is recommended.

Following the suggestion, we have conducted western blot analysis of KDM2A and ACTB in whole-cell lysates prepared from sgCtrl or sgK#1-transduced ALT#1 cells. The new data is now included as Figure 6c.

p. 17, line 386: Moreover, SMC5 siRNA treatment in sgK#1-transduced ALT#1 cells significantly prevented the abnormal M-phased multitelomere cluster formation as compared to those treated with scrambled control siRNA (Fig. 7a-c).

Comment: According to Fig. 7a, the localization of SMC5 was changed by KDM2A KO, though the amount of SMC5 protein was not affected (Fig 7c). Did KDM2A affect AMC5 localization?

After careful evaluation of all relevant SMC5/TelC immuno-FISH images, we found no consistent pattern of SMC localization change in KDM2A-depleted cells compared to the control cells. We therefore conclude that KDM2A depletion does not significantly affect SMC5 localization in ALT cells. We apologize for any confusion and have replaced the previous image with a more representative one in Figure 7c.

p. 18, line 388: Moreover, SMC5 siRNA treatment in sgK#1-transduced ALT#1 cells significantly prevented the abnormal M-phased multitelomere cluster formation as compared to those treated with scrambled control siRNA (Fig. 7a-c). These results indicate that KDM2 indeed acts downstream of SMC5/6.

Comment: Because RNAi knock-down of SMC5 blocked the recruitment of telomeres to PML bodies and significantly reduced multitelomere cluster foci formation (Supplementary Fig. 7a, b), reduction of the abnormal M-phased multitelomere cluster formation would simply reflect the reduced levels of multitelomere cluster foci formation by SMC5 KD. Indeed, the difference between the values of siCtrl and siSMC5 in sgCtrl cells seemed to be similar and the difference between the values of siCtrl and siSMC5 in sgK#1 cells.

We totally agree with the reviewer. RNAi knock-down of SMC5 blocked multitelomere cluster formation in ALT cells and consequently reduced their need for KDM2A-facilitated mitotic telomere declustering.

p. 19, line 432: Analysis of the G2-synchronized ALT#1 cells found that the Flag-tagged SENP6C1030A mutant was indeed localized to clustered telomeres (Fig. 7k).

Comment: In sgCtrl cells, the dotted Flag signals were detected, most of which were overlapped with telomere foci. In the sgK#1 cells, there was one circle that was not overlapped with telomere foci. Was this circle overlapped with a signal for PML? In the Figure 5b PML were detected as dots overlapping telomere signals in sgCtrl and sgK#1 cells. If not, did the expression of SENP6C1030A mutant disrupt PML in the context of

KDM2A KO ALT1 cells? Please detect PML in Figure 7k. It is also recommended to detect KDM2A protein in Fig. 7k.

The reviewer raised an interesting point. Following the suggestion, we have repeated the experiment and conducted immuno-FISH analysis of telomere, Flag-SEN6^{C1030A} and PML co-localization in sgCtrl or sgK#1-transduced ALT#1 cells expressing Flag-SEN6^{C1030A}. The data show that expression of SEN6^{C1030A} in KDM2A-depleted ALT1 cells did not significantly affect PML association with telomeres. The new analyses are included to replace the old ones in Figure 7k and i. In addition, a western blot analysis of KDM2A expression has been incorporated into Figure 7j as suggested.

When they claim the direct involvement of KDM2A in ALT, it is required to show if SENP-6 directly binds to KDM2A or SENP-6 directly binds to H3K36, depending on the methylation status.

As suggested by the reviewer, we have conducted a co-immunoprecipitation experiment to examine whether SENP6 physically binds to KDM2A. Western blot analysis of the samples did not uncover evidence of physical interaction between KDM2A and Flag-SEN6. The new data is included as Supplementary Figure 16.

With regard to investigating the possibility of physical interaction of SENP6 with telomere H3K36 of different methylation status, we found it difficult if not impossible to design such an in vivo experiment in which that we can separate and distinguish telomere nucleosomes based on their H3K36 methylation status. We feel that a clean experiment will likely require in vitro biochemical assay. As SENP6 binds to SUMOylated substrates, the in vitro assay will need to examine its interaction with reconstituted recombinant telomere nucleosomes containing combinations of H3K36 methylation and SUMOylated histones. But because histone SUMOylation has been identified at many lysine residues of H2A, H2B, H3 and H4, a rigorous exploration of the impact of H3K36 methylation on SENP6-mediated histone de-SUMOylation would represent a large-scale study requiring a very significant amount of time. We therefore hope that the reviewer will agree that is beyond the scope of current study.

p. 22, line 501; p.23, line 512-521: Our results further indicate that the process is regulated by KDM2A-directed telomere histone H3K36me2 demethylation.

Comment: There was few evidences showing that KDM2A-directed telomere histone H3K36me2 demethylation functions in ALT. The results in Figure 3 showed that KDM2A demethylase activity and DNA binding activity were involved in the ALT process. These activities may be involved in controlling the expressions of the genes required for ALT de-clustering process. Since SENP6 binds to SUMOylated substrates,

The reviewer raised a very interesting point. While our data may support a direct involvement of KDM2A in ALT-mediated telomere maintenance through facilitating mitotic telomere declustering, we agree with the reviewer that we cannot totally exclude the other possibility that KDM2A may contribute indirectly through regulating the expression of genes involved in ALT. We have added discussion into the revised manuscript as suggested by the reviewer.

Reviewer #3 - ALT in cancer, CRISPR screens (Remarks to the Author):

I read the study by Li et al, "Histone demethylase KDM2A is a selective vulnerability of cancers relying on alternative telomere maintenance" with great interest. In brief, the authors used a targeted screening approach to identify that KDMA2 deletions results in specific lethality of ALT cells. Through directed molecular biology, they identify loss of KDMA2 in ALT cells impairs the ability of cells to detangle the phase-separated telomere clusters (i.e. APBs) that are known to occur in cells of this type. They identified KDMA2 functions upstream of SENP6, which is required to deSUMOylate proteins within APBs. Loss of KDMA prevents SENP6 recruitment, and the continued entanglement of presumed recombination intermediates in mitosis leads to mitotic catastrophe.

Overall, the study is of excellent quality and represents a novel approach to identifying putative vulnerabilities in ALT cancers. I particularly appreciated the use of varied and well-designed CRISPR screens. Supporting experiments were largely well done and informative to the study. I find the report strong and believe it is an excellent candidate for publication once the authors address the concerns I have below. I stress these are largely minor concerns and expect they can be handled by the research team. Congratulations to the authors on an excellent study.

We appreciate this reviewer's positive assessment and encouragement.

1. Line 125: How did the authors identify 1,000 cells per sgRNA coverage?

The average input coverage was determined by dividing the number of infected cells with the number of sgRNAs within the library. To maintain the representation of sgRNAs, the number of infected cells during the screen was kept at least 1,000 times the sgRNA number in the library. In the screens of the chromatin regulator-associated library (consisting of 5014 sgRNAs), the numbers of infected cells were then kept at a minimum of 5.5×10^6 .

2. Line 140: Describing the spiked in controls would help orient the reader.

We have added the description to the text as suggested.

3. Line 336: Is there a reference for cell-cycle-dependent PML-NB or APB dissociation?

The reference (Draskovic et al., PNAS 2019; 106: 15726-15731) has been added into the revised manuscript as suggested.

4. Line 353: This line is somewhat misleading. TRF1 does not dissociate from mitotic telomeres and can readily be seen on mitotic telomeres (e.g. with GFP-TRF1). However these telomere foci are dim compared to the very bright APBs being visualized. This should be re-worded to better convey.

We have corrected this in the text as suggested.

5. Mitotic catastrophe is a very poorly defined phenomena. What specifically is occurring? Soley chromosome segregation errors, or is it also mitotic death (a known consequence of lethal replication stress, PMID: 31530811).

Thanks for the insightful comments. Following the suggestion, we went back to analyze the mitotic outcomes of the control and KDM2A-KO ALT#1 cells in our live-cell imaging data. The analysis revealed a strong mitotic death phenotype in the KDM2A-depleted ALT cells. Moreover, our western blot analysis of the control and KDM2A-KO IMR90-T and ALT#1 cells showed an increased level of cleaved PARP1 specifically in KDM2A-depleted ALT#1 cells, suggesting an apoptosis-dependent mitotic death. These new data are now included into the revised manuscript as Supplementary Figure S10e-g.

6. Discussion: The description of "synthetic lethality" within the ALT context is difficult as KDM2A is not synthetic lethal with a second gene but is instead 'synthetic lethal' with a phenotype (i.e. ALT). The Authors show indeed KDM2A is not synthetic lethal with ATRX. IT might be worth clarifying for the reader.

Thanks for pointing it out. We have re-worded the text to make it clear in the revised manuscript.

7. Figure 2 – these data would benefit by tarking KDM2A in matched cancer types of the ALT-cancer cells used in this study. I.e. a telomerase positive osteosarcoma line (to match with U2OS and Saos2).

Following the reviewer's suggestion, we have conducted the competition-based proliferation assay in telomerase positive osteosarcoma cell line MG63. The new data is now included in Supplementary Figure S4a.

8. The movies, while lovely to watch, lack significant and important details. There are no time stamps to indicate time, there are no scale bars to indicate size. Further it is not clear which events are important. The authors need to include time stamps and scale bars, and it may be better to identify individual events with arrows or other markers to draw the readers attention.

Following the suggestion, we have now added time stamps, scale bars, and square markers into the live imaging movies.

Minor comments

9. The imaging related to telomere condensates being maintained in mitosis in the KDM2A is striking. While not critical to this study, should the authors be able to visualize this with live imaging it would be a powerful tool.

We totally agree with the reviewer. Actually, we have been trying to knock-in a mCherry-tag into the endogenous TRF1 loci, but so far have not been successful. And we do not know whether the signal and resolution will be strong enough for live imaging. But nevertheless, we will give it a try.

Reviewers' Comments:

Reviewer #1:

Remarks to the Author:

The authors have done a fine job. The manuscript is much improved and the data relating to the dependency on KDM is good and important. Nice job!!

The link with SEMP6 remains somewhat unclear. I had hoped that this would be clarified in greater detail - which I believe adds more mechanistic insights in the study.

Reviewer #2:

Remarks to the Author:

The specific activation of ALT in tumors makes it a potential target for therapy. However, there are currently no successful target therapy. In this paper, they developed isogenic pairs of ALT cell lines from human lung fibroblast, and performed CRISPR-based pooled genetic screens with protocols minimizing the noises. They identified KDM2A, which was specifically required for survival of ALT-dependent cells. Further, they showed that KDM2A was required for disassembly of ALT-specific recombinogenic multitelomere bodies after homology-directed telomere DNA synthesis. Finally, they suggested that this de-clustering was mediated through KDM2A-controlling the multitelomere localization of SENP6 that executed SUMO deconjugation of telomere-associated proteins. Their strategy to identify KDM2A as a target for ALT gives this manuscript high originality. The experiment provided by the authors support their conclusions, which present important clues for therapy targeting ALT. I suggest that the paper is suitable for publication.

Reviewer #3:

Remarks to the Author:

Thank you to the authors for addressing all the reviewers concerns. I was unable to find supplementary movie legends, which need to be added before final publication. All other concerns were addressed. I recommend publication.

Detailed response to recommendations

We are grateful to the reviewers for their support and appreciate the opportunity to submit a revised manuscript.

Reviewer #1

The authors have done a fine job. The manuscript is much improved and the data relating to the dependency on KDM is good and important. Nice job!!

We thank the reviewer for the compliment.

The link with SEMP6 remains somewhat unclear. I had hoped that this would be clarified in greater detail - which I believe adds more mechanistic insights in the study.

We recognize the reviewer's point. Our co-immunoprecipitation experiment revealed that SENP6 does not physically interact with KDM2A (Supplementary Figure 16a). In addition, complementation of wild-type but not KDM2A mutant defective of demethylase (D214A) restored SENP6^{C1030A} recruitment to ALT telomeres in sgK#1-transduced ALT#1 cells (Supplementary Fig. 15a-c), suggesting that KDM2A indirectly regulates SENP6 recruitment to ALT telomeres, likely through modulating telomere H3K36 methylation and interaction with H3K36me2 readers. Since this is a new field with many knowledge gaps, to understand how KDM2A regulates ALT-directed telomere maintenance and SENP6 recruitment, we will need to conduct many large-scale telomere proteomic and following-up verification studies like the ones pioneered by O'Sullivan group (Garcia-Exposito et al., 2016, Cell Rep 17:1858). Those studies are of great interest to us but will require years of painstaking work. We therefore hope the reviewer will agree that the raised point could be addressed separately from the current manuscript, as it already consists of significant amount of original work focusing on identifying KDM2A as a target for ALT,

Reviewer #2

The specific activation of ALT in tumors makes it a potential target for therapy. However, there are currently no successful target therapy. In this paper, they developed isogenic pairs of ALT cell lines from human lung fibroblast, and performed CRISPR-based pooled genetic screens with protocols minimizing the noises. They identified KDM2A, which was specifically required for survival of ALT-dependent cells. Further, they showed that KDM2A was required for disassembly of ALT-specific recombinogenic multitelomere bodies after homology-directed telomere DNA synthesis. Finally, they suggested that this de-clustering was mediated through KDM2A-controlling the multitelomere localization of SENP6 that executed SUMO deconjugation of telomere-associated proteins. Their strategy to identify KDM2A as a target for ALT gives this manuscript high originality. The experiment provided by the authors support their conclusions, which present important clues for therapy targeting ALT. I suggest that the paper is suitable for publication.

We thank the reviewer's positive assessment and support.

Reviewer #3

Thank you to the authors for addressing all the reviewers concerns. I was unable to find supplementary movie legends, which need to be added before final publication. All other concerns were addressed. I recommend publication.

We appreciate this reviewer's support. The supplementary movie legends have now been included into the Supplementary information.